

# 1 Modelling nutrient retention in the coastal zone of an

# 2 eutrophic sea- a model study

*Elin Almroth-Rosell[*1], Moa Edman[1], Kari Eilola[1], H.E. Markus Meier[2,1], Jörgen Sahlberg[1]*
[1]Swedish Meteorological and Hydrological Institute, Norrköping, Sweden
[2]Leibniz Institute for Baltic Sea Research Warnemünde, Rostock, Germany
[*]**Corresponding author:**
Elin Almroth-Rosell
Swedish Meteorological and Hydrological Institute
Sven Källfelts gata 15, SE-426 71 Västra Frölunda, Sweden.
Tel: +46(0)31 7518969. E-mail: elin.almroth.rosell@smhi.se
**Keywords:** Nutrient retention, phosphate, nitrogen, eutrophication, biogeochemistry, coastal
zone, Baltic Sea

# 16 Abstract

The Swedish Coastal zone Model (SCM) was used at a test site, the Stockholm Archipelago
located in the northern part of the central Baltic Sea, to study the capacity of the coastal filter
on nitrogen (N) and phosphorus (P). The SCM system is based on the Swedish Coastal and
Ocean Biogeochemical model (SCOBI) coupled to the equation solver PROgram for
Boundary layers in the Environment (PROBE). In this study the Stockholm Archipelago
consisting of 86 sub-basins was divided into three sub-areas: the inner, the intermediate and
the outer archipelago. An evaluation of model results to observations showed that the
modelled freshwater supply and the nutrient, salinity and temperature dynamics in the SCM
model are of good quality. Further, the analysis showed that the Stockholm Archipelago
works as a filter for nutrients that enter the coastal zone from land, but the filter capacity is
not effective enough to take care of all the nutrients. However, at least 65 % and 72 % of the
P and N, respectively, are retained. Highest total amounts of P and N are retained in the outer





archipelago where the surface area is largest. The area weighted specific retention of P and N,
however, is highest in the smaller inner archipelago and decreases towards the open sea. A
major part of the retention is permanent, which for P means burial. For N almost 92 % of the
permanent retention is represented by benthic denitrification, less than 8 % by burial, while
pelagic denitrification is below 1 %. A reduction scenario of the land loads of N and P
showed that the retention capacities of N and P increase and the export of N from the
archipelago decreases. About 15 years after the reduction the export of P changes into an
import of P from the open sea to the archipelago.



## 1 Introduction

The worldwide increase of coastal eutrophication and anoxia has spread exponentially since the 1960s. Coastal oxygen depletion is associated with dense population areas and large river loads of nutrients (Diaz and Rosenberg, 2008). The use of industrially produced fertilizer started in the late 1940s and has since then been contributing to the anthropogenic fertilization of the global marine system (Galloway et al., 2008). The river load of nutrients originating from agriculture activities has been shown to be controlled by the size of the river flow, e.g. the flow from the Mississippi River has a large impact on the oxygen conditions in the northern Gulf of Mexico, which suffers from severe hypoxia with "dead zones" as a result (Rabalais et al., 2002). The enhanced land load of nutrients to the Baltic Sea (Fig. 1) located in northern Europe, during the period 1950-1980 (Gustafsson et al., 2012) has led to eutrophication and consequently increased frequency and intensity of cyanobacterial blooms, expanding bottom hypoxia and dead bottom zones (e.g. Bergström et al., 2001; Conley et al., 2009; Diaz and Rosenberg, 2008; Vahtera et al., 2007). Actually, the largest anthropogenically induced hypoxic area in the world is found in the Baltic Sea (Carstensen et al., 2014), where it varied between 70000 and 80000 $km^2$ during year 2010-2014 (Hansson and Andersson, 2014).

With ambition to diminish eutrophication there has been a lot of efforts around the world to reduce the land load of nutrients to sea, but the expected results of a healthier environment has not been accomplished in all places (Kemp et al., 2009). In e.g. the Baltic Sea, most of the coastal zones and the open sea still suffers from eutrophication in spite of reduced nutrient loads (HELCOM, 2010). The response of eutrophication and the extent of hypoxic area for changes in nutrient loads is different in different types of systems and also changes in climatic and hydrodynamic conditions might lead to a non-linear recovery (Kemp et al., 2009). Nutrients transported from land to sea first enter the coastal zones and are then further transported towards the open sea. However, not all of the supplied nutrients reach the open sea as the coastal zone acts as a filter (McGlathery et al., 2007). Nutrients are involved in coastal chemical, physical and/or biological processes (e.g. denitrification, burial, algae and plant assimilation) and are, thus, retained in coastal areas (Duarte and Cebrián, 1996; Voss et al., 2005). The retention capacity of the coastal zone might be of large importance for the water quality in open waters like in the eutrophic Baltic Sea. On the other hand, open waters might also affect the coastal filter and its retention of nutrients by the exchange of nutrients from the open waters to the coastal zone (Humborg et al., 2003).





Retention capacity is, however, not well defined. Johnston (1991) discussed that retention
processes are of different magnitudes and irreversibility, e.g. plant uptake and litter
decomposition provide short- to long-term retention of nutrients, which depends on release
rates, translocations, and the longevity of the plants. Billen et al. (2011) and Nixon et al.
(1996) defined retention as the net effect of temporary and permanent removal from the water
phase through different biogeochemical processes (e.g. biological uptake, burial and
denitrification). The net effect of nutrients on the water quality of an area can be studied by
the simple method of subtracting the output of nutrients from the input (Johnston, 1991). This
simple method of calculating the retention capacity of nitrogen (N) and phosphorus (P) has
been used in a number of studies (e.g. Eilola et al., 2014; Hayn et al., 2014; Karlsson et al.,
2010; Nixon et al., 1996; Sanders et al., 1997) for different areas of the world. The retention
capacity has been discussed to be related to the residence time and depth in different water
systems (Balls, 1994; Hayn et al., 2014; Nixon et al., 1996).  Hence, the longer a water parcel
and its nutrient content stays within a system, the larger the containing nutrients are affected
by the internal transformation and retention processes.
There are not enough estimates of nutrient retention in different coastal zones around the
world to evaluate and understand its effect on the environmental status of coastal seas. Thus,
quantification of the retention capacities in different coastal ecosystems as river outlets,
archipelagos, lagoons and embayments would increase the understanding and the knowledge
necessary for managing the coastal zone. The aim with this study is to quantify the retention
capacity of N and P and to discuss the relative importance of different physical and/or
biological processes in the coastal archipelago of the Swedish capital Stockholm, using the
Swedish Coastal zone Model (SCM). The Stockholm Archipelago is the largest archipelago in
Sweden and the second largest in the Baltic Sea. Signs of eutrophication in the Stockholm
Archipelago have been observed as increased ratio of laminated sediments from the 1930s
(Jonsson et al., 2003) and the eutrophication status in the inner Stockholm Archipelago were
in the early 1970s classified as highly eutrophic (Lännergren et al., 2009).
An implemented P reduction in the sewage treatment facilities in the 1970s led to some
improvements of the marine environment (Brattberg, 1986), but in the 1990s the areas were
still eutrophic with poor bottom water oxygen conditions (Jonsson et al., 2003; Rosenberg and
Diaz, 1993). The reduction of the N load from the sewage treatment facilities in the mid-
1990s led to further improvement of the eutrophication status. In 2008 the bottom oxygen
conditions had clearly improved in the deeper parts and only enclosed bays, such as e.g. Stora





Värtan, suffered still from anoxia (Karlsson et al., 2010 and references therein). However, the
annual monitoring status report of the environmental status of the inner Stockholm
Archipelago in 2014 still classified the area as unsatisfactory eutrophic (Lücke, 2015).
The large river Norrström supplies high nutrient loads to the inner part of the archipelago why
this area is suitable for retention calculations. Changes in the retention capacity along the
land-sea continuum, from the inner archipelago, through the intermediate and outer
archipelago to the open Baltic Sea will be studied in order to evaluate the effect of the size of
the archipelago on the retention capacity.
After a description of the model system (Section 2) and an evaluation of the results of SCM
(Section 3.1), the retention capacity of the coastal zone is calculated and the effects of a
reduced land load of N and P is analyzed (Section 3.2). Conclusions finalize the study
(Section 4).



## 2 Methods

### 2.1 Study site

The brackish archipelago of Stockholm (Fig. 1), located at the east coast of Sweden, is a continuation of the river Norrström with an average discharge of about 160 m$^3$ s$^{-1}$ from Lake Mälaren (Lindh, 2013). The river outflow carries about 2600 metric tons (t) of N and 120 t of P annually to the coastal basin "Strömmen" in the inner archipelago (Lännergren, 2010). The rocky islands in the archipelago are surrounded by basins of different sizes and depths which are connected by straits. In this study the archipelago has been divided into three areas: the inner, intermediate and outer archipelagos. Several large islands form a natural border between the inner and the intermediate archipelagos and the limited water exchange occurs through five narrow sounds with shallow sills. The outflow from the inner to the intermediate archipelago passes through the sounds in the surface layer, while inflows of more saline water mainly occur at larger depths. The border between the intermediate and the outer archipelagos follows the chain of islands in north-south direction with several connections between the areas (Fig. 1).

The largest point sources of nutrients to the inner archipelago originate from waste water treatment facilities of Stockholm, which is situated at the outlet of the Lake Mälaren. The chemical and biological treatment removing P was implemented in the early 1970s and was further developed in the 1990s to remove also N.

*Fig. 1.*

### 2.2 Model description

The Swedish Coastal zone Model (SCM) is a multi-basin 1D-model based on the equation solver PROgram for Boundary layers in the Environment (PROBE;Svensson, 1998), coupled to the Swedish Coastal and Ocean Biogeochemical model (SCOBI; Eilola et al., 2009; Marmefelt et al., 1999). The model system was developed to calculate physical and biogeochemical states in Swedish coastal waters. The inner, intermediate and outer Stockholm archipelagoes (Fig. 1) are represented by 16, 44 and 26 sub-basins, respectively (see figure in Supplement 1).



**2.2.1 PROBE**
The physical model PROBE calculates horizontal velocities, temperature and salinity profiles
(Svensson, 1998). The surface mixing is calculated by a $k$-$\varepsilon$ turbulence model and the bottom
mixing is a parameterization based on the stability in the bottom water. Light transmission, as
well as ice formation growth and decay, are also included in the model. The vertical grid
resolution is half a meter in the uppermost layers, one metre from 4-70 m, and two metres
between 70-100 m. The general differential equation of the PROBE solver is formally written
as
$$\frac{\partial \phi}{\partial t} + \frac{\partial}{\partial x_i} u_i \phi = \frac{\partial}{\partial z}\left(\Gamma_\phi \frac{\partial \phi}{\partial z}\right) + S_\phi \qquad (1)$$
Here $\phi$ is the dependent variable, $t$ time, $z$ vertical coordinate, $x_i$ horizontal coordinates, $u_i$
horizontal velocities, $\Gamma_\phi$ vertical exchange coefficient, and $S_\phi$ source and sink terms. Vertical
advection (and moving surface) is included accounting for vertical transport in sub-basins due
to in and outflows. The sources and sinks determined by the ecosystem model are added to $S_\phi$.
The water exchange between the sub-basins is controlled by the baroclinic pressure gradients.
The net flow through the sounds will be the same as the river discharge from land in order to
preserve volume. Inflowing water to a sub-basin is interleaved into its density level without
any entrainment, and heavy surface water in one sub-basin may thus reach the bottom level in
an adjacent basin. The sea level variations outside the boundary are of minor importance for
the SCM results and are therefore not included in the forcing. The water exchange across the
boundary between the coastal zone and the open sea is assumed to be in geostrophic balance,
since this boundary is open with a width greater than the internal Rossby radius.
**2.2.2 Biogeochemical model (SCOBI)**
The SCOBI model describes the biogeochemistry of marine waters in the Baltic Sea and
Kattegat (Eilola et al., 2009). Nine pelagic and two benthic variables (Fig. 2) are described in
the SCM-SCOBI model. In the pelagic zone three different phytoplankton groups (diatoms,
flagellates and others, and cyanobacteria), one zooplankton group, one pool for detritus and
three inorganic nutrients pools (nitrate, ammonium and phosphate) are represented. The
model also calculates oxygen and hydrogen sulfide concentrations, of which the latter are
represented by "negative oxygen" equivalents (1 ml $H_2S$ $l^{-1}$ = $-2$ ml $O_2$ $l^{-1}$) (Fonselius and



Valderrama, 2003). For the benthic layer the amounts of stored N and P are calculated.
SCOBI has been used and validated in several studies, both coupled to the basin scale Baltic
Sea model PROBE-Baltic (Marmefelt et al., 1999) and to the three dimensional Rossby
Center Ocean model (RCO; e.g. Meier et al., 2011).
In the model the processes of phytoplankton assimilation, mortality and nitrogen fixation,
zooplankton grazing, excretion of detritus and dissolved inorganic nitrogen (DIN) and
phosphorus (DIP), the oxygen dependent mineralization of detritus, benthic N and benthic P,
nitrification and denitrification are described. Phytoplankton assimilates carbon (C), N and P
according to the Redfield molar ratio (C:N:P=106:16:1) and the biomass is represented by
chlorophyll (Chl) according to a constant carbon to chlorophyll mass (mg) ratio (C:Chl=50:1).
Light attenuation depends on background attenuation due to water and humic substances and
a variable attenuation caused by particulate organic matter (phytoplankton, zooplankton and
detritus). All particulate variables sink downward through the water column. Predation is used
as a closing term to parameterize interactions with higher tropic levels in the ecosystem and
move matter from zooplankton to the detrital and inorganic pools. Resuspension of sediment
that is important in the open Baltic Sea (Almroth-Rosell et al., 2011) has not yet been
implemented in this SCOBI version, but the sediment releases dissolved inorganic nutrients
back to the water mass, with the release of phosphate being redox dependent. Some fractions
of benthic N and P are assumed to be buried in the sediment as a permanent sink, and are
hence removed from the system. For further details of the SCOBI model the reader is referred
to Eilola et al. (2011; 2009).
*Fig. 2.*

### 2.2.3   Forcing

The SCM-SCOBI model system is forced by weather, atmospheric deposition of nutrients, the
conditions in the sea outside the archipelago, point sources, and discharge of freshwater and
nutrients from land. The initial values for both the pelagic zone and the sediment are derived
from spin-up periods of the model.
There are two types of land derived forcing; discharge of water and nutrients from both rivers
and surface run-off from the drainage area given by the S-HYPE model (Lindstrom et al.,
2010) and point sources representing sewage plants and industries. The run-off is added to the
surface water of each basin and no reduction of river nutrients due to precipitation at river-
mouths is assumed in this model setup. The point sources of nutrient loads are assigned to the





depth levels most resembling the actual depth of the discharge. Organic nutrients in the land
load are calculated from the difference between total nitrogen (TN) and DIN, and total
phosphorus (TP) and DIP, respectively. The organic N and P that is in balance according to
Redfield ratio will be added to the detritus pool in the model, while the remaining part is
regarded as not bioactive dissolved organic nutrients.
The weather forcing consists of solar insolation, air temperature, wind, relative humidity and
cloudiness. The insolation and all the radiation and heat fluxes across the water-air interface
are calculated by the PROBE model. The weather variables are taken from a gridded database
developed at the Swedish Meteorological and Hydrological Institute (SMHI), using 3-hourly
meteorological synoptic monitoring station data, and the depositions of nitrogen species
(NHX and NOX) are calculated by the MATCH model (Robertson et al., 1999). For the
deposition of phosphate, a literature value of 0.5 kg m$^{-2}$ month$^{-1}$ is used (Areskoug, 1993).
The boundary conditions to the open Baltic Sea is set by vertical mean profiles calculated by a
one dimensional PROBE setup for each Baltic open water area and assimilation of monitoring
data. The monitoring data used in the assimilation are extracted from the different stations
MS4, US5B, SR5, BY31 and BY29 (Fig. 1) depending on depth and time, to get the best
representation of the open sea's influence on the SCM model domain.
**2.3 Evaluation strategy**
To quantify the fit between modelled values and observations a correlation coefficient, $r$, was
calculated (Eq. 2).

$$r = \frac{\sum_{i=1}^{n}\left(P_i - \overline{P}\right)\left(O_i - \overline{O}\right)}{\pm\sqrt{\sum_{i=1}^{n}\left(P_i - \overline{P}\right)^2 \sum_{i=1}^{n}\left(O_i - \overline{O}\right)^2}}$$

where $P$ is model value, $O$ is observation of the analyzed parameter, $i$ is the data number and $n$
is the total number of data points. Two series of observations and model values that are
identical will lead to an $r$ value equal to one, while uncorrelated data result in a $r$ value close
to zero. In addition to the $r$ value, the average cost function (C) values (Eq. 3) for the different
parameters were used in the evaluation of the SCM results.





$$C = \dfrac{\sum\limits_{i=1}^{n} \left| \dfrac{P_i - O_i}{sd(O_i)} \right|}{n}$$                3
A cost function describes the proximity of model results and observations by normalizing the
difference between them with the standard deviation (sd) of the observations. If average
model results fall within the standard deviation of observations, C is below one which is
regarded as good. Results that are within two standard deviations will be regarded as to be on
an acceptable level. The corresponding simulation levels, good and acceptable, for the
correlation coefficient are achieved when r is higher than two thirds (0.66) and one third
(0.33), respectively. This approach using *r* and C has been used in earlier studies (Edman and
Omstedt, 2013; Edman and Anderson, 2014) and is based on methods by Oschlies (2010).
The outflow from Lake Mälaren is three orders of magnitude larger than the sum of all other
S-HYPE fresh water components to the inner Stockholm Archipelago. The output from S-
HYPE of fresh water and nutrient loads from Mälaren to the Stockholm Archipelago was
therefore used in the evaluation of the fresh water forcing to SCM. Observations of freshwater
discharge were retrieved from the Baltic Environmental Database (BED, 2015) at the Baltic
Nest Institute, Stockholm University. The *r* value between monthly mean of observed and
simulated discharge for the period (1990-2012) was then calculated.
In the evaluation of the results of the SCM in different basins, the long term averages (1990-
2012) of the vertical distribution of salinity, DIN, DIP and oxygen during winter (November-
February) and summer months (May-August) were compared to corresponding observations
for the whole modelled period. Further, the correlation *r* and the mean cost function C of the
vertical distribution of observations and model output were calculated. Also the long term
averages of the seasonal variations in surface temperature, DIN, DIP and bottom water
oxygen concentrations were used in the evaluation by calculating the corresponding *r* and C
values.
Observations from the Stockholm Archipelago (Fig. 3) were provided by Stockholm City and
Stockholm University. For the quantitative validation described above the quality of
observations from each site (Table 1) had to fulfil three requirements to be used in the
validation process; 1) period coverage: 80% of the years sampled; 2) annual coverage: at least
7 of the 12 months sampled; and 3) vertical data coverage: at least 5 depth levels frequently
measured over the full depth of the basin. In addition at least 3 months with observations were
required for the evaluation of winter and summer conditions. Average values were then




calculated for periods and depth levels with dense data distribution. The model output was
used in the same way as observations, and the modelled averages were calculated for the same
time intervals and depth ranges.
*Fig. 3.*
*Table 1.*

## 2.4  Calculation of retention

The retention of P and N in a region can be calculated as the difference between the load and
the outflow (Almroth-Rosell et al., 2015; Hayn et al., 2014; Johnston, 1991; Meier et al.,
2012). The input of nutrients is the sum of inflows from outer areas, rivers, land runoff, point
sources and atmospheric load, while the outflow of nutrients is the export from the area to
outer seas (Fig. 4). In the present study the focus has been on the retention of the external
nutrient load from land and atmosphere. The different retention processes have been
calculated separately, as they are included in the biogeochemical model SCOBI. Total
retention ($R_{TOT}$) is the sum of both permanently and temporally retained P and N. Burial is the
only retention process for modelled P that permanently removes P from the model system,
while for N also benthic and pelagic denitrification has to be considered as permanent sinks.
A build-up of the pelagic and benthic active pools of N and P are regarded as temporal
retention processes. $N_2$-fixation is another process that needs to be taken into account as it is a
source of bioavailable N to the system. The retention (or sink) efficiency, $R_{TOT}$ (%), of the
sum of nutrient load from land and deposition from air ($Nu_{in}$) has been calculated according to
eq. 4:
$$R_{TOT}(\%) = 100 \times \frac{R_{TOT}}{Nu_{in}} \qquad\qquad 4$$
The total retention and retention efficiency were calculated for the entire Stockholm
Archipelago but also separately for the inner, intermediate and outer archipelagos in order to
investigate the spatial gradient of retention capacity from the inner coastal zone towards the
open Baltic Sea.
*Fig. 4.*





## 3 Results and discussion

### 3.1 Validation

The load of freshwater to the model is an important forcing component, which needs to be of good quality since the coastal zone is strongly influenced by the land load of freshwater and nutrients. The variability of the modelled discharge of water and nutrients by the S-HYPE model is compared to observations for the simulation period (1990-2012) with good results (Fig. 5 and Table 2). A clear relationship between the magnitude of river outflow ($Q_F$) and the nutrient loads is observed both for monthly observations and S-HYPE output (Fig. 5). The model seems to slightly underestimate the spring discharge and overestimate low flow regimes. However, overall it captures a realistic annual variation of the discharge, which is reflected in high correlation coefficients (Table 2) for all evaluated parameters. Highest correlation coefficients are found for $Q_F$ and TN, compared to the slightly lower values for TP, DIP and DIN, which is in accordance with previous studies (Grimvall et al., 2014; Strömqvist et al., 2012). An extensive validation is also available in Sahlberg et al. (2008).

*Fig. 5.*

*Table 2.*

Datasets from eight stations (Table 1) fulfilled the requirements of good data availability and were used in the evaluation of the SCM model results. There are aspects that are important to have in mind when comparing model results and observations. The model characteristics are horizontally averaged described, while an area as the Stockholm Archipelago has relatively large spatial salinity gradients. The representativeness of the station is therefore somewhat limited when compared to model results. Observations may in general also be influenced by local conditions, e.g. sewage effluents, high sediment fluxes or stagnant conditions, which are smeared out in the average results of the model. Still we assume for the present study that the station data are good enough for the quantitative model validation and give a background for discussions about model strengths and weaknesses. As an example, validation results are shown for the innermost basin Strömmen, which is one of the basins where the number of observations is large enough both during summer and winter periods to be included in the validation process. The observations are from station Blockhusudden (Position G in Fig. 3) situated at the boundary between basin Strömmen and the next adjacent sub-basin.





The long term average summer depth profiles of modelled salinity and oxygen in the basin
Strömmen correlate well with observations, while the winter values of salinity were too low.
Especially in the surface layers (Fig. 6a,b). This difference is partly due to the fact that the
salinity of a station at the entrance to the basin is more reflecting the boundary conditions of
the downstream basin than the mean conditions in Strömmen. The surface winter
concentrations of oxygen were too high, but decreased with depth and became too low in the
lower layers (Fig. 6b). The reason for these discrepancies is unclear but the results fall within
the large variability of observations. However, it might be expected that the winter surface
oxygen concentrations in observations should be higher than in the summer because of the
temperature effect on oxygen saturation concentrations. The results indicate that there is an
impact from local conditions at the monitoring station that are not captured by the model
setup. The modelled DIN depth profiles show higher values at about 15 m depth during both
winter and summer (Fig. 6c), while the DIP profiles values seems to be satisfactory at all
depth and periods (Fig. 6d). Also the individual observations show higher concentrations of
both DIN and DIP around 15 m depth which is where the halocline has its largest vertical
gradient. This depth level corresponds to the depth where two sewage water treatment plants
relieve their sewage water in the model. The winter stratification was stronger in the model
because of the lower surface salinity. This hamper the vertical transports of oxygen and has an
influence on the winter oxygen conditions in the deep water that were lower in the model
compared to the observations from the more well ventilated entrance area.
The average seasonal variation of the surface temperature and the bottom water oxygen
concentrations was captured by the model, but not the increase of nutrients, especially DIN,
during autumn (Fig. 7). The surface salinity was overall somewhat low, which probably is a
result the location of the monitoring station, described above.
The objective correlation coefficients and the cost function value for the different parameters
implied correspondingly that the model managed to simulate the average vertical winter and
summer profiles with good or acceptable skills in the basin Strömmen (Fig. 8g), except for the
average seasonal value of DIN that was described as not good. The differences between
model results and observations of  DIN may be due to similar reasons as described for salinity
above. In the other basins used in the evaluation (vertical and seasonal profiles are not shown)
of the SCM all parameters during winter, summer and season were simulated with good or
acceptable skills, except for the average vertical summer profiles of DIN in the basin
Solöfjärden (Fig. 8c) and oxygen concentration in the basin Sandöfjärden (Fig. 8a). The





combined model skills, which were calculated as the average of the individual *r* and C values,
were good in six of the eight evaluated basins (purple cross in Fig. 8). In the remaining two
basins the skills were considered as acceptable.
*Fig. 6.*
*Fig. 7.*
*Fig. 8.*
**3.2   Retention of nutrients in the Stockholm Archipelago**
During the period 1990-2012 on average 174 t P $yr^{-1}$ and 5846 t N $yr^{-1}$ entered the inner
archipelago, mainly from the Lake Mälaren. That is a major part of the 217 t P $yr^{-1}$ and
8288 t N $yr^{-1}$ which entered the entire Stockholm Archipelago (Fig. 9). The P load from point
sources was clearly lower than the river load (Fig. 10). However, the N load from point
sources was higher than the river load in the beginning of the studied period in the inner
archipelago (Fig. 10 d), but decreased in the middle of the 90s due to the implementation of a
more effective method to remove N in the waste water treatment facilities. The P supply to the
intermediate archipelago mainly originated from runoff from land (Fig. 10 b), while for N
there were also some point sources that contributed to the land load on the same level (Fig. 10
e). In the outer archipelago the nutrient load from land was almost negligible and most of the
nutrients were deposited from the atmosphere (Fig. 10 c, f).
*Fig. 9.*
*Fig. 10.*
Largest amounts of P and N were retained in the outer archipelago compared to the
intermediate and inner archipelago (Fig. 9). The retention of all supplied P and N, including
the net import from upstream areas were within the inner, intermediate and outer Stockholm
archipelago 18 %, 23 % and 48 % for P, respectively, and 14 %, 26 % and 60 % for N,
respectively. The area of the three zones increases from inner (109 $km^2$), to the intermediate
(759 $km^2$) and to the outer archipelago (2360 $km^2$) and thus, the retention of nutrients seems
to increase with increased area. On the other hand, the amount of retained P and N per area
unit ($km^{-2}$) was highest in the inner archipelago, and decreased towards the open sea (Fig. 11).
The water depth and the residence time are affecting the retention of nutrients, which will be
further discussed in Section 3.2.2. The largest part of the total retention in the entire
Stockholm Archipelago was permanent, which for P means burial. For N benthic





denitrification represented as much as almost 92 % of the permanent retention, burial for less
than 8 % and pelagic denitrification was below 1 %.
*Fig. 11.*
In the inner Stockholm Archipelago about 14 % and 18 % of the external N and P supply,
respectively, were retained within the area. Karlsson et al. (2010) found in their empirical
study for 1982-2007 that about 15 % of the total input of N and 10 to13% of the total input of
P were retained in the inner Stockholm Archipelago. However, their numbers are based on the
total input, thus both the land load and an estimated input from outer areas, i.e. the
intermediate Stockholm Archipelago. After a recalculation from the given numbers in their
study about 24 and 25 % of the N and about 21 and 29 % of the P load from land were
retained within the system for the periods 1982-1995 and 1996-2007, respectively. These
numbers of the retention efficiency are higher than the numbers in the present model study. A
recalculation of the retention efficiency in the SCM for the latter period (1996-2007) in the
inner archipelago did not change the SCM results considerably. The largest difference
between the two studies is caused by the calculation of net exchange of nutrients through the
sounds. The transport through the sounds was in Karlsson et al. (2010) calculated from
average volume flows estimated from mass balance calculations for salt together with budget
calculations using observations of average nutrient concentrations. In the present study the
exchange of nutrients between the inner and the intermediate archipelago was part of the
dynamic model calculations in the SCM.
The temporary retention in SCM is negative in all three parts of the archipelago for both P and
N (Fig. 9). The reason for negative temporary retention is mainly a decrease in the benthic
nutrient pools during the period (Fig. 12). The largest decrease (29 %) is found in the pelagic
pool of N in the inner archipelago, which coincides with the decrease in N load from point
sources (Fig. 10). In the intermediate and outer Stockholm archipelagos the pelagic pool of N
remains on about the same level through the whole simulation period. The large decreases in
the benthic pools of N and P (14-18 %) occur in the intermediate and outer archipelagos,
while there are only small changes in the pelagic and benthic pools of P in the inner
archipelago. Because of the nutrient retention there is a reduced net transport of N and P from
the inner archipelago towards the intermediate and outer archipelagos and further to the open
sea during the period (Fig. 9). The importance of the import of nutrients into the coastal zones
from sea has been discussed in earlier studies (e.g. Humborg et al., 2003). However, the effect





of nutrient transports from the open sea into the coastal zone on coastal retention has not been
studied before and will be addressed in more detail in future studies.
*Fig. 12.*

### 3.2.1  The coastal filter

From the present results it can be concluded that the Stockholm Archipelago works like a
filter for nutrients that enter the coastal zone from land. However, a rather large area of the
archipelago is needed to effectively retain the nutrients. About 82 and 86 % of P and N
supplies, respectively, pass the small inner archipelago and are exported to the intermediate
archipelago. In the intermediate and the outer archipelago all local supplies of nutrients from
land and atmosphere are retained together with a fraction of the nutrients imported from the
inner archipelago. The efficiency of the total retention increased with the increased coastal
area from land to the sea continuum (Fig. 13). However, the filter capacity of the entire
Stockholm Archipelago is not effective enough to take care of all the nutrients that enter the
system from land and the atmosphere, but still, at least 65 % and 72 % of the supplied P and
N, respectively, are retained. The total retention numbers (permanent and temporary)
correspond to 141 t P yr$^{-1}$ and 5954 t N yr$^{-1}$ (Fig. 9). Since Stockholm Archipelago is the
largest archipelago in Sweden it might be that most of the other Swedish coastal areas with a
large run-off from land would be less effective as coastal filters and, thus, contribute to a
larger extent to the eutrophication in the open sea. This is one question in focus of an on-
going study where the entire Swedish coastal area will be evaluated similarly to the present
study.
*Fig. 13.*

### 3.2.2  Processes affecting retention

The present study was performed in an area characterised as an eutrophic archipelago in an
inland sea with basins having oxic, hypoxic and anoxic bottom waters. The results of the
retention are in agreement with results from previous studies (Billen et al., 2011; Hayn et al.,
2014; Nixon et al., 1996), but with somewhat lower values in the entire and inner plus
intermediate archipelagos (Fig. 14). Their studies were performed in various types of systems:
coastal lagoons, drowned river estuaries, coastal embayments, and inland seas in North
America and in Europe. Those systems varied from being relatively pristine to systems with
large point sources (eutrophic), and they also varied between oxic to hypoxic and/or anoxic
conditions. Nixon et al. (1996) showed that the retention of P and N correlated to the log scale



of the ratio between the average depth and the residence time of the study areas, which is
confirmed by the results from the present study (Fig. 14). However, it is not clear for all the
mentioned studies above how the residence time was calculated, why it is possible that the
results are not completely comparable. In the present study the residence time is calculated as
the volume of the study area divided by the fresh water inflow from land. In shallow areas
larger parts of the sinking particulate organic material may reach all the way down to the sea
floor where it can be exposed to retention processes as burial and denitrification. On the other
hand, in a much deeper area a larger part of the organic material may become re-mineralised
within the water column on its way down to the sea floor. The nutrients can then be re-used
by phytoplankton and/or be further transported out from the system. Long residence times in a
system increase the time of exposure for biogeochemical transformation processes and
sedimentation within the system and larger parts of the nutrients may be retained. These two
parameters therefore affect the retention of nutrients (Fig. 14) and the correlation implies that
the retention occurs mostly in the sediment due to processes such as burial and denitrification
which is the case in the present model study. Benthic primary producers and benthic fauna are
also important factors for the retention of nutrients in shallow coastal ecosystems
(McGlathery et al., 2007; Norkko et al., 2012). These processes are not yet implemented in
the SCM, and therefore not included in the present study. It is also important to know whether
a system is in balance with the nutrient loads or not since it would affect the retention
capacity. In this study the temporary retention is negative for both N and P in all three areas of
the Stockholm Archipelago which implies that the system is not in a steady state. This
imbalance is however expected since there are reductions of the nutrient loads in the first part
of the simulation period (Fig. 10 a, d). However, the possibility that the results may be
influenced by unknown initial conditions of sediment concentrations should not be excluded.
There are only few observations available and the knowledge about the amount of sediment
nutrients involved in biogeochemical cycles is poor.
*Fig. 14.*

### 3.2.3   Reduction scenario of the nutrient land load

The SCM is used to investigate the effect of a reduction of the nutrient load from land to the
Stockholm Archipelago. The loads of P and N from sewage treatment facilities depend on
their size, i.e. the number of person equivalents (pe) for which they were built. Estimates of
realistic minimum discharge concentrations of P and N from sewage treatment facilities used
in the SCM reduction scenario, which are based on technical feasibility but not on economic



or resource sustainability, are given in Table 4 (Kerstin Rosén Nilsson, County
Administrative Board of Stockholm, personal communication). The reductions of P and N
from land, e.g., due to decreased nutrient load from agriculture and increased use of small
sized sewage treatment plants by individual households where these are not connected to a
municipal sewage treatment facility are set to 10 % and 15 % for P and N, respectively. Also
the point sources from different industries are assumed to decrease their discharge of N and P
with 10 % as well. The reductions result in a total decrease of P and N load with 12 and 20 %,
respectively, to the entire Stockholm Archipelago. Largest impact from the load changes
occurs in the inner archipelago, where most of the nutrient load enters the sea.
A SCM model spin-up run period of 45 years with river and weather forcing from year 2010
provides the steady state initial conditions used for the reduction experiment. After the spin-
up period the reductions of the nutrient loads are implemented, according to Table 4.
*Table 3.*
Fastest response is seen in the pelagic pool of N which rapidly decreases, but reaches a steady
state after about three years with reduced loads (Fig. 15). The pelagic pool of P decreases in
the inner archipelago, but increases slightly in the outer areas. The changes in P pools are
slower compared to those in N pools. The large and fast decrease of pelagic N in the inner
archipelago, results in a decreased N:P ratio (Table 3), as well as (not shown) lower
chlorophyll concentrations, reduced sedimentation, and increased export of P from the inner
archipelago to the outer areas and the Baltic proper. Also the anoxic areas decrease as a result
of the lower deposition of organic material on the sea floor (not shown). The changes in the
benthic pools of N and P occur over a longer time period and the benthic P pool does not
reach a steady state before about 40 years after the reduction.
In the reduction scenario the transport of N to the open sea from the Stockholm Archipelago
decreases by 62% within four years (Table 3). The retention capacity of N in the entire
archipelago increases at the same time from 79 % to 90 % as a result of the load reduction.
The longer response time of P compared to N is observed also in the retention capacity (Fig.
15).The retention efficiency of P is in the end of the spin-up run about 100 %. This implies
that the under the conditions during year 2010 all the total P load is retained in the Stockholm
Archipelago when the system is in steady state. The retention of P, however, then decreases to
74 % during the first years after the reduction, coinciding with the large decrease in the N





pelagic pool and the decrease in N/P ratio. After the initial decrease, the retention capacity
slowly increases to 105% at the end of the simulation period, i.e. retention is larger than the
land and atmospheric load of P. As a consequence the export of P from the archipelago to the
Baltic proper decreases with time, and about 18 years after the load reduction the direction of
the transport changes. This coincides with the time when the retention capacity again reached
100 %. Thereafter the archipelago instead begins to import P from the open sea. Thus, with
the contemporary boundary conditions used at the open sea, P from the Baltic proper becomes
retained within the archipelago. For coastal management this indicates the importance of the
open sea nutrient conditions because of the interactions with the nutrient cycling coastal zone.
*Table 4.*
*Fig. 15.*



## 4 Conclusion

Archipelagos are complex areas with many basins and several shallow sounds, which affect the transport of water and the dissolved and particulate nutrients. For the first time the SCM model was used to study the capacity of the coastal filter of nutrients. An evaluation showed that model results are of good quality compared to observations.

We focused our study to the northern Baltic proper and investigated retention of N and P in the Stockholm Archipelago. The main findings are described below.

- The coastal zone works as an efficient filter for the land loads of nutrients. Under prevailing conditions 65 % and 72 % of P and N supplied from land, respectively, are retained.

- A sensitivity experiment reducing the land load of nutrients showed that the retention capacity of N and P increased. In this case the export of N from the archipelago decreased and P was imported from the open sea.

- The average filter capacity is dependent on the spatial dimensions of the coastal area. Nutrient retention per area is largest in the inner archipelago and decreases towards the open sea.

- Average water depth and water residence time regulate the retention of nutrients that occurs mostly in the sediment due to processes such as burial and denitrification.

- The pools of nutrients in the water and in the sediment changes with nutrient loads on different time scales and affects the temporal nutrient retention in the area. N has a rather short response time of about three years while it takes about 40 years for P to reach balance in a system with constant forcing. Changing N:P ratios in the archipelago due to the different response time scales also have an impact on the nutrient retention capacity on decadal time scales.

- Coastal management needs to take the aspects of time and balance between nutrient loads and pools into account in the assessment of impacts from nutrient load abatements. On shorter timescales the retention capacity might seem less effective when the nutrient load from land decreases.

 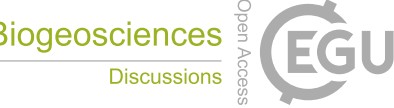

## 1  **5  Acknowledgement**

The research presented in this study is part of the Baltic Earth programme (Earth System
Science for the Baltic Sea region, see http://www.baltex-research.eu/balticearth) and is part of
the BONUS COCOA (Nutrient COcktails in COAstal zones of the Baltic Sea) project which
has received funding from BONUS, the joint Baltic Sea research and development
programme (Art 185), funded jointly from the European Union´s Seventh Programme for
research, technological development and demonstration and from the Swedish Research
Council for Environment, Agricultural Sciences and Spatial Planning (FORMAS), grant no.
2013-2056. Additional funding came from the EU Water Framework Directive programme at
the Swedish Meteorological and Hydrological Institute. We would also like to thank Kerstin
Rosén Nilsson at the County Administrative Board of Stockholm for interesting discussions
and good advice.



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



## 7  Figures

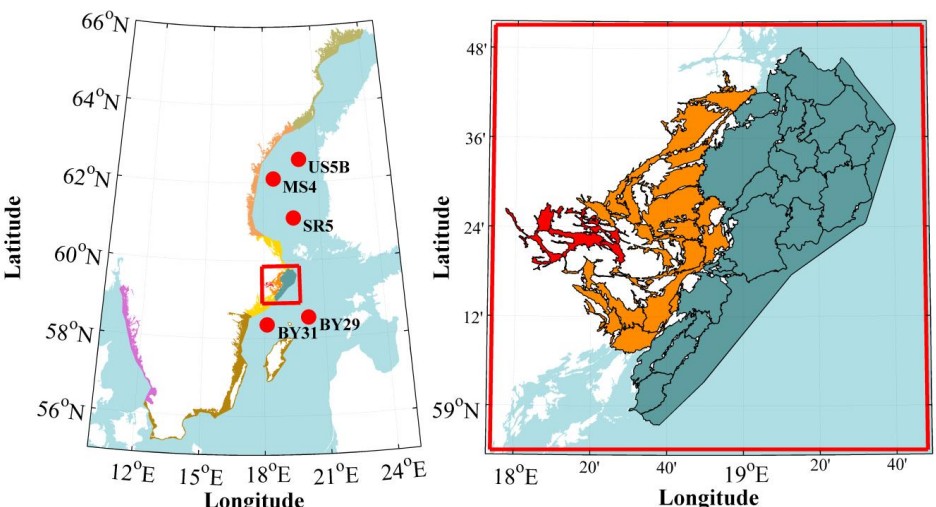

Fig. 1.



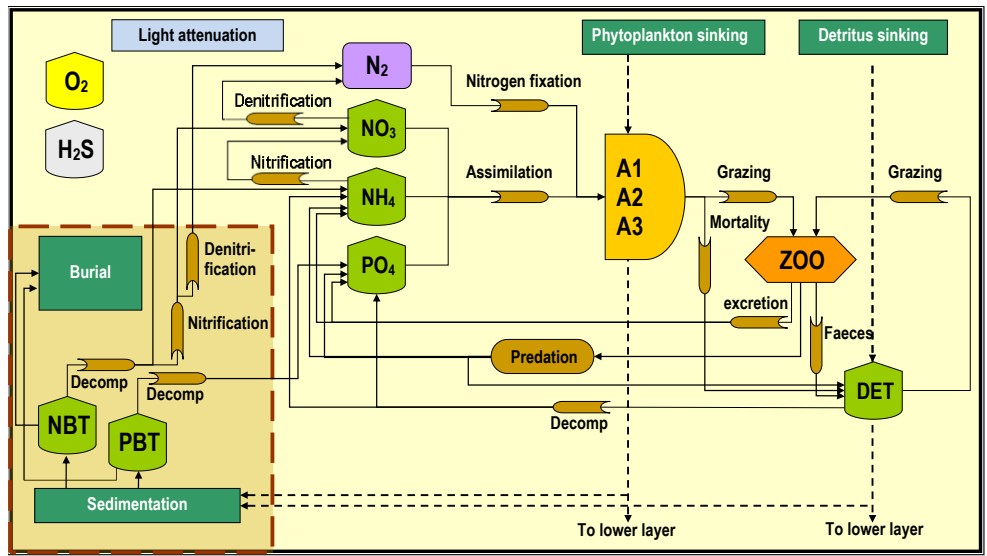

Fig.2.





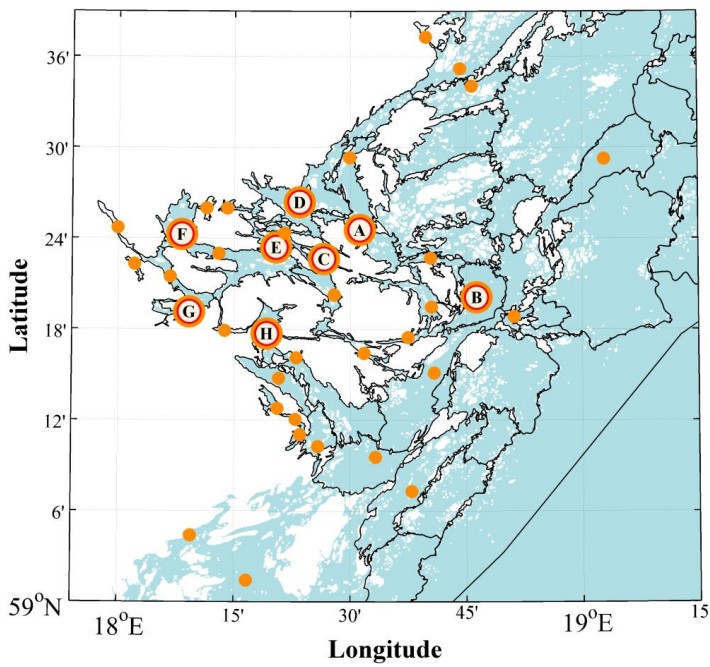

Fig. 3.



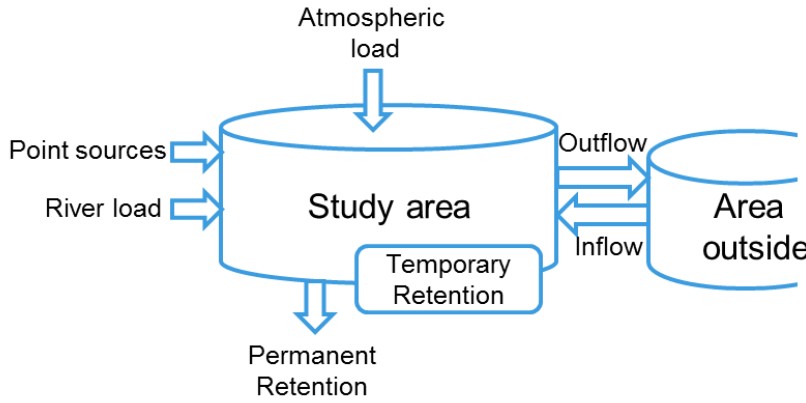

Fig. 4.





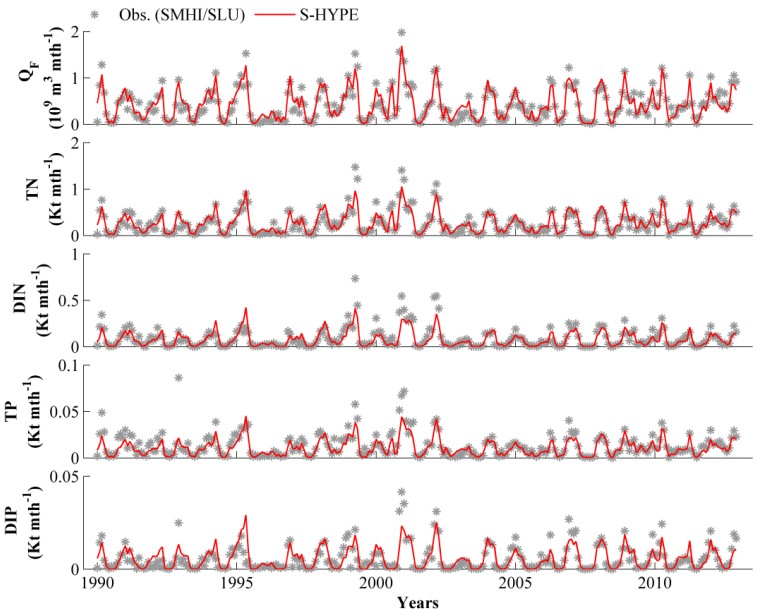

Fig. 5.





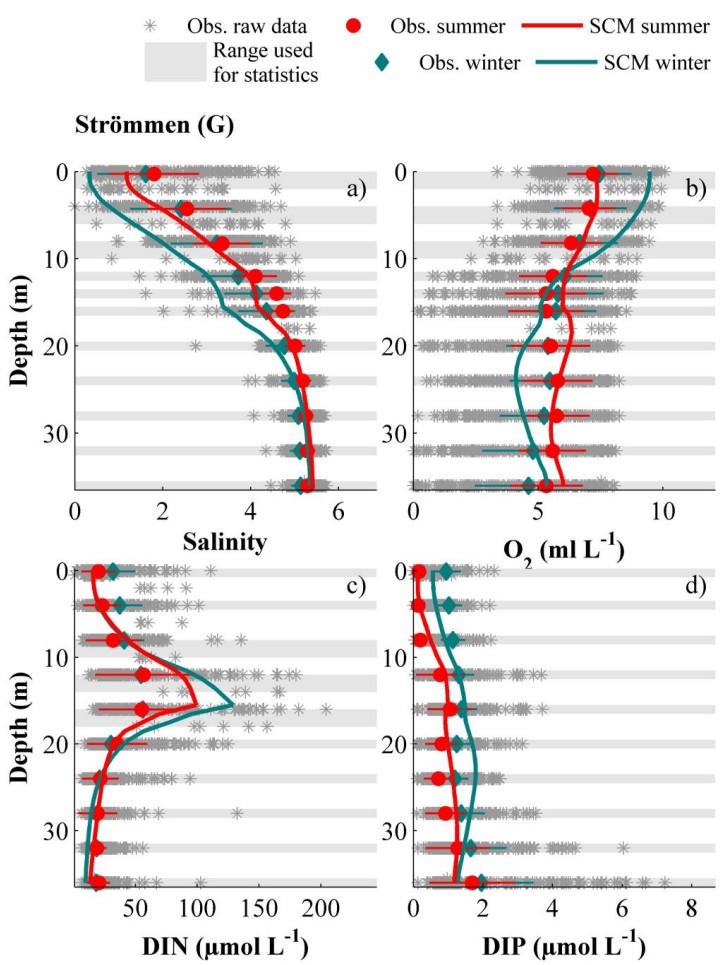

Fig. 6.





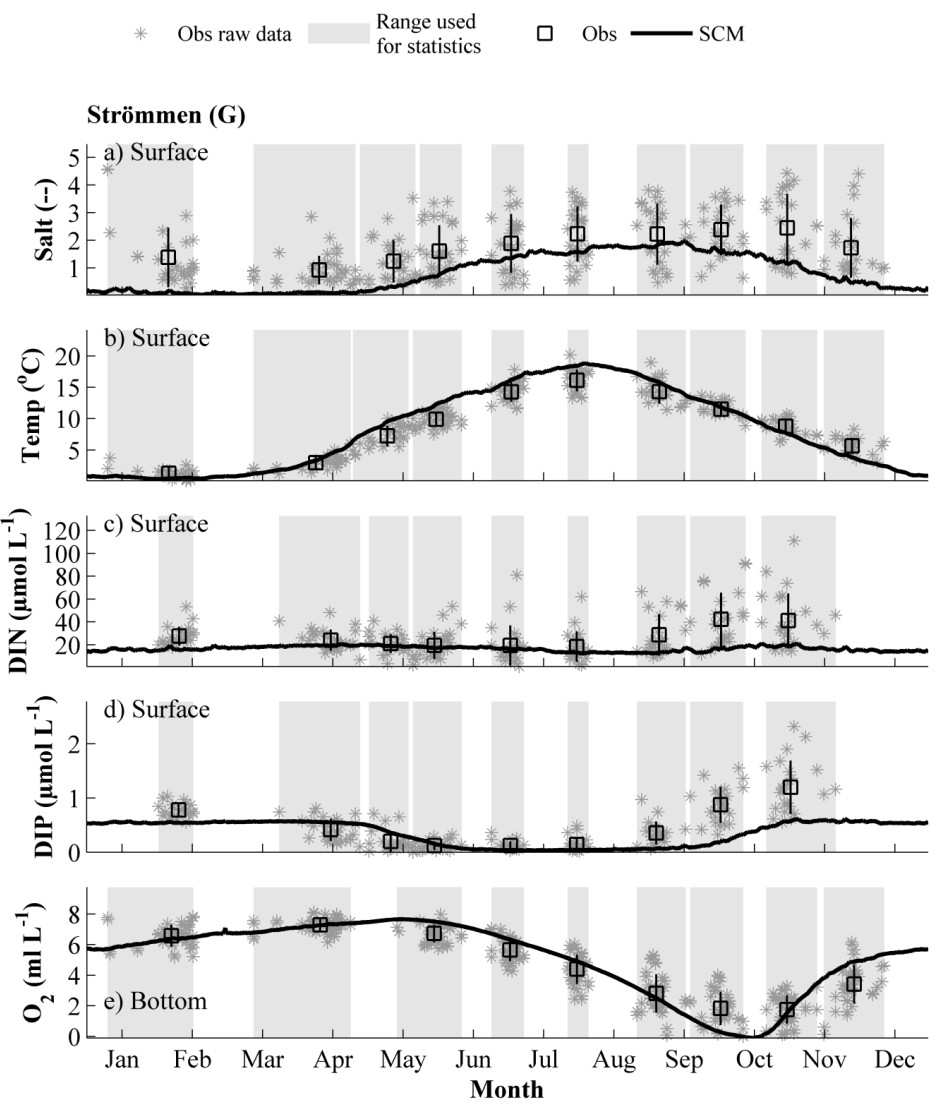

Fig. 7.





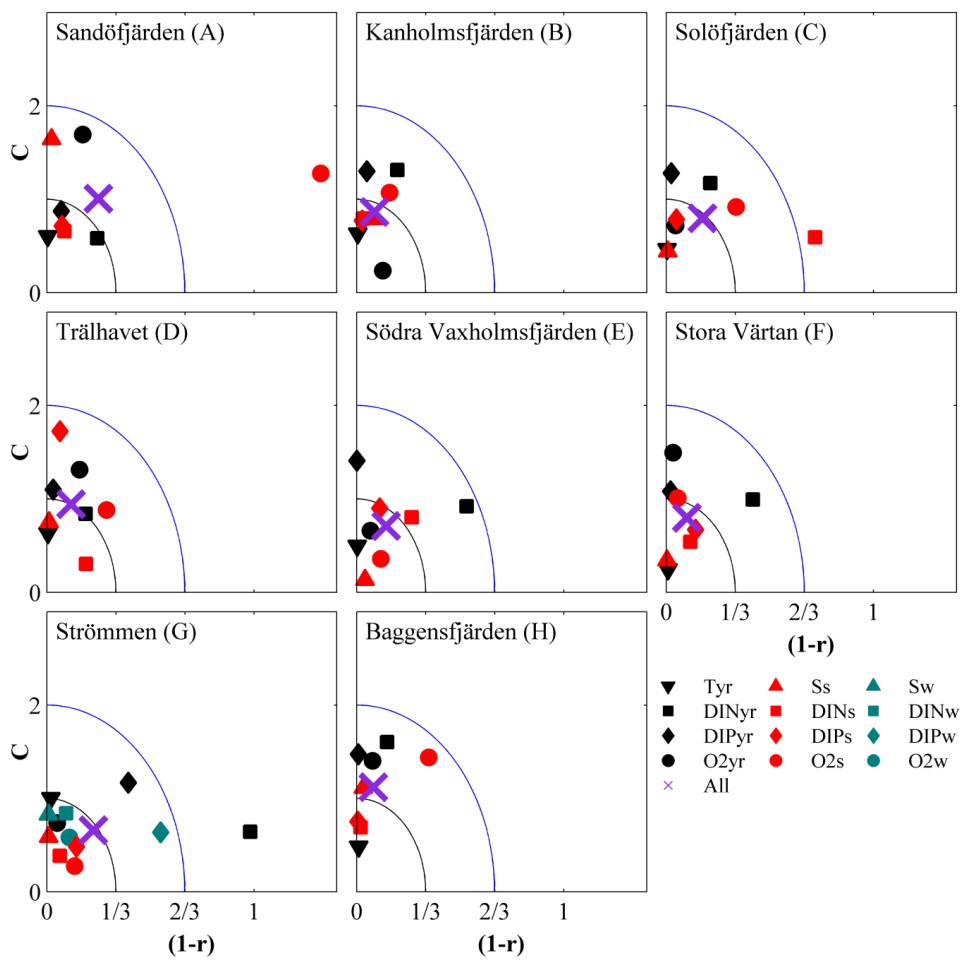

Fig. 8.

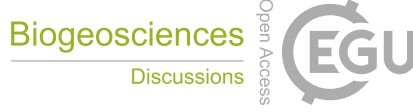



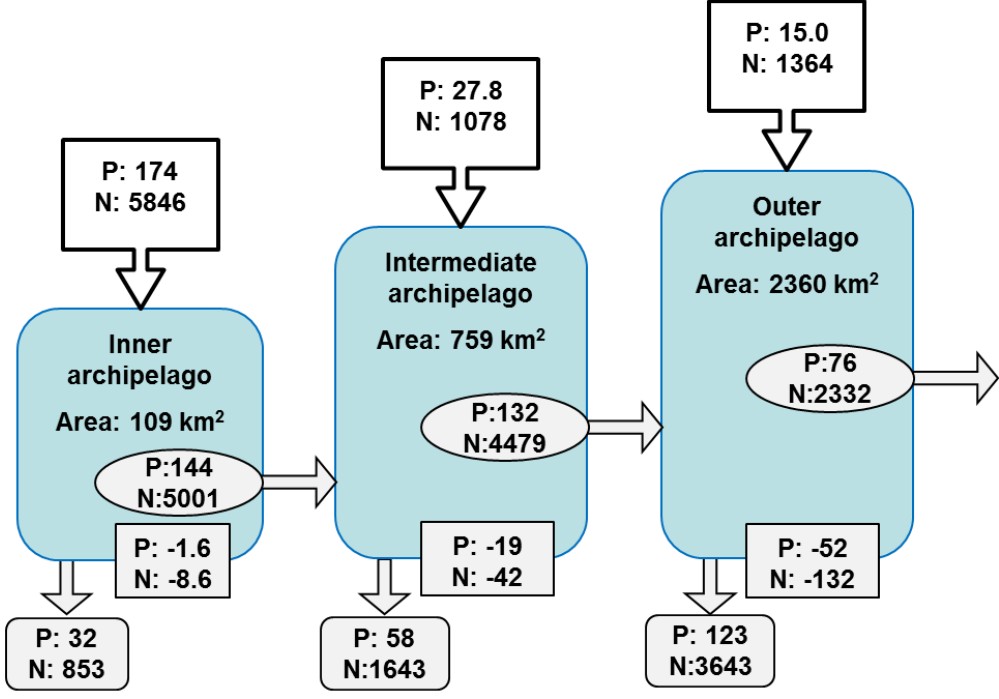

Fig. 9.



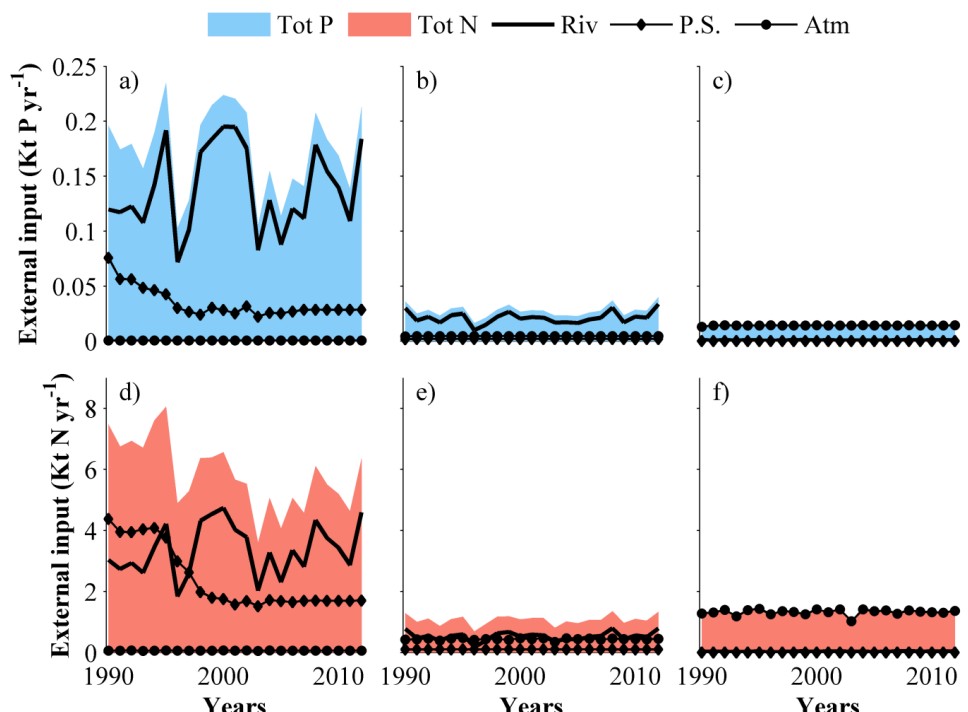

Fig. 10.





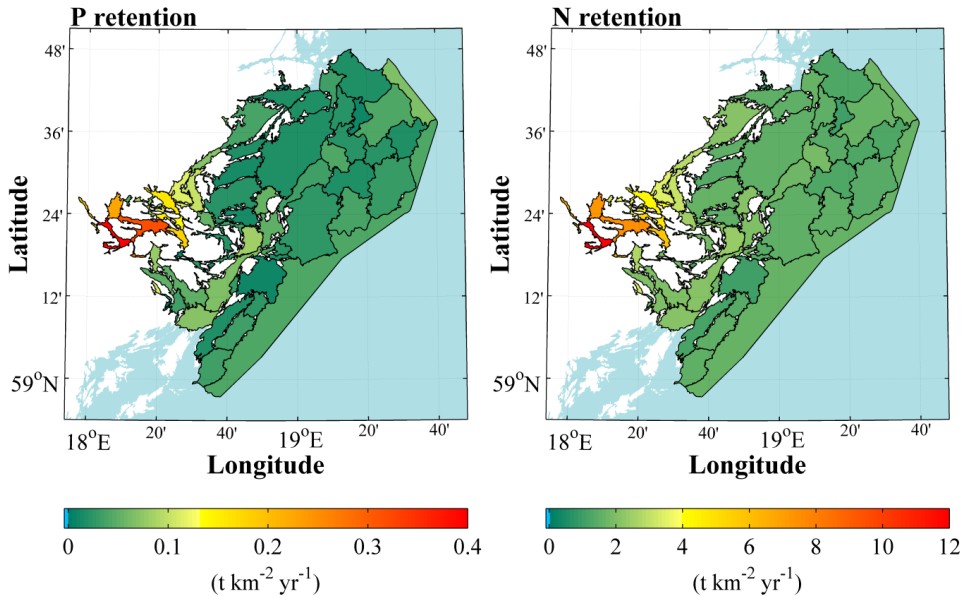

Fig. 11.



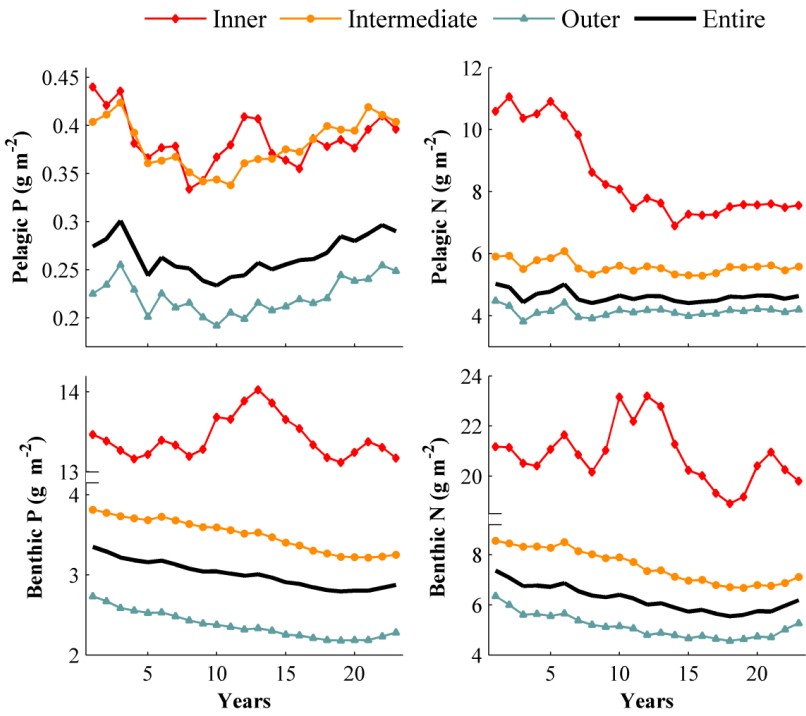

Fig. 12.

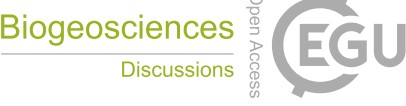

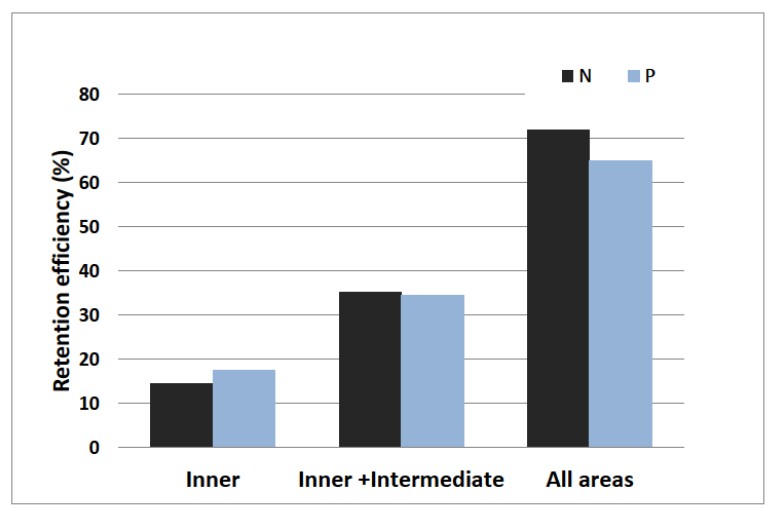

Fig. 13.



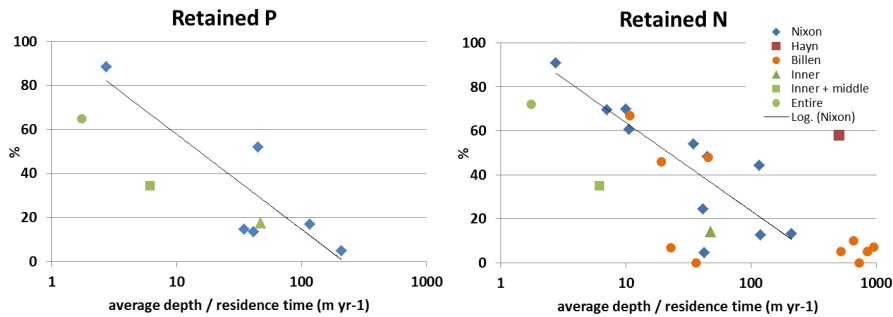

Fig. 14.





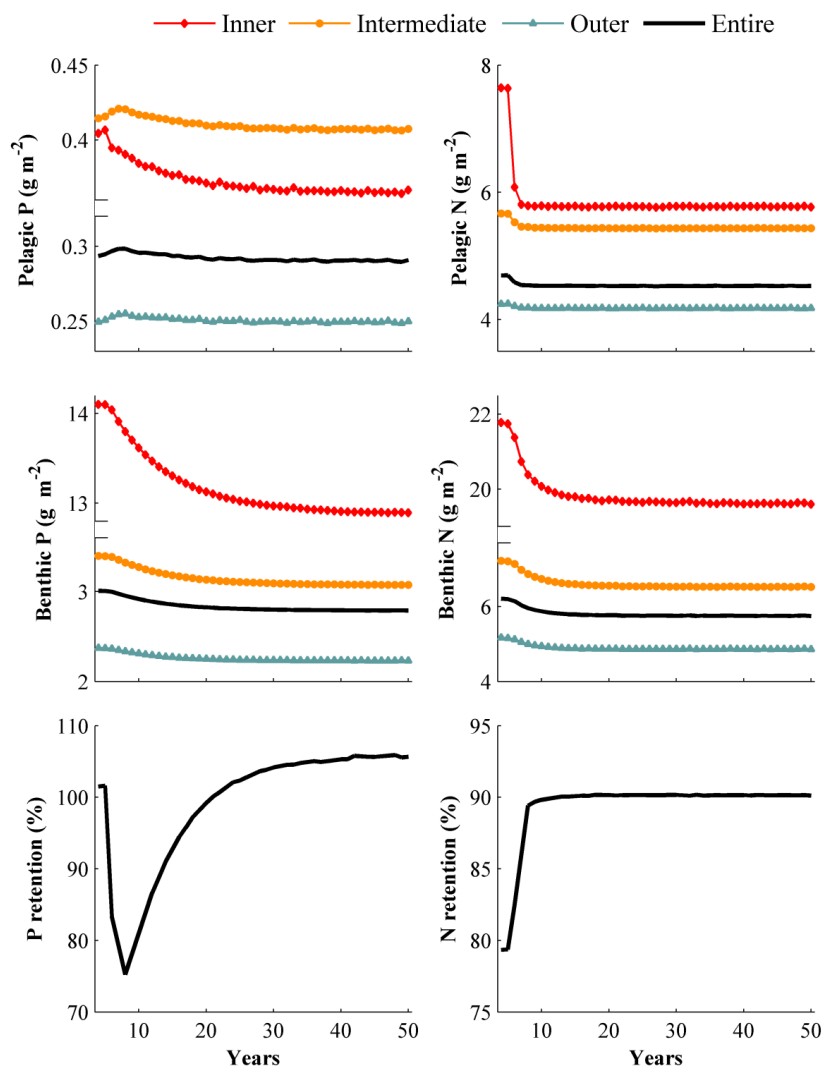

Fig. 15





## 8 Figure captions

*Fig.1. The Swedish Coastal zone Model can be used in different areas along the Swedish coast streching from the Norwegian border in the West to the Finnish border in the North (different colours, left). In the present study the SCM model covers the northen Baltic proper (marked with a red square) and has been used to estimate the coastal retention capacity of nutrients in the Stockholm inner (red), intermediate (orange) and outer (blue) archipelagos (right).The outlet of river Norrström is marked with a black arrow and the different basins are shown by the black contours.*

*Fig.2. Schematic figure of the Swedish COastal and BIogeochemical model, SCOBI. Oxygen and hydrogen sulphide are simplified for clarity.*

*Fig.3. Available locations with observations (circles and dots) in the Stockholm Archipelago. Model evaluation of temperature, salinity, DIN, DIP and bottom water oxygen concentration was performed at selected stations (circles marked with letters), which are described in Table 1.*

*Fig.4. The calculation scheme of retention in the study area.*

*Fig.5. Observed (stars) and modelled (line) monthly outflow ($Q_F$) and nutrient loads from Lake Mälaren, through Norrström to basin Strömmen for the modelled period (1990-2012). DIN is the sum of nitrate and ammonium.*

*Fig.6. The SCM modelled (lines) and observed (circle and diamond) vertical average profiles (1990-2012) of salinity (a) and concentrations of oxygen ($O_2$; b), DIN (c) and DIP (d) in the basin Strömmen during winter (turquoise) and summer (red) months. Depth layers with dense number of observations (grey stars) determined the vertical depth intervals (grey shaded area) used in the profile calculations. The standard deviations (horizontal lines) were calculated for the summer and winter values of the observations.*

*Fig.7. Simulated (lines) and observed averages (squares) of the seasonal variation and the standard deviaton (vertical lines) of the observations in the basin Strömmen (1990-2012) of surface temperature (Temp), salinity, DIN and DIP and of the bottom water oxygen concentrations. Time periods with dense number of observations (grey stars) determined the time intervals (grey shaded area) used in the calculations.*

*Fig.8. Average cost function (C) and correlation coefficients, adjusted (1-r) to the range 0-1, for an overview of the model skill at the eight different validation sites (A-G). The individual skills of the different parameters, average seasonal variation (black) and/or the vertical summer (red) and winter (turquoise) profiles of temperature (T), DIN, DIP and oxygen concentrations (O2) are shown, as well as the combined model skills for all variables (purple cross). Variables within the inner quarter circle and between the two quarter circles are considered to be good and acceptable, respectively, while variables that are outside the quarter circles are not well simulated.*

*Fig.9. Transport scheme of N and P (t yr$^{-1}$) from land and atmosphere (top boxes), and the net exchange from the inner, intermediate and outer archipelago (ellipse) towards the open sea. Total retention is the sum of temporary retention (square) and permanent retention (square with round corners). For P burial is the only process that leads to permanent retention, while for N also denitrification removes N. Negative values for the temporary retention means a decrease in the benthic and/or pelagic pools of nutrients.*

*Fig.10. The annual total P and N (coloured area) load (t yr$^{-1}$) and the contributions from rivers and land (solid line), point sources (diamonds) and atmosphere (filled circles) to the inner (a, d), intermediate (b, e) and outer (c, f) Stockholm Archipelago during the period 1990-2012.*





*Fig.11. The retention per area unit (t km$^{-2}$ yr$^{-1}$) of P (left) and N (right) in each basin of the Stockholm Archipelago.*

*Fig.12. The total content (g m$^{-2}$) of the pelagic (top) and benthic (bottom) P (left) and N (right) in the inner (diamonds), intermediate (circles), outer (triangles) and entire (black line) Stockholm archipelagos.*

*Fig.13. The retention efficiency (%) of P (blue) and N (black) from river, land run-off and atmosphere in the inner archipelago towards the open sea by adding the next area and its input to the calculation.*

*Fig.14. The retention of P (left) and N (right) versus the logarithmic ratio between the average depth and the residence time of the study areas (m yr$^{-1}$). Data from other studies are from Billen et al. (2011); (Hayn et al., 2014; Nixon et al., 1996). The straight line shows the logarithmic regression for the data from Nixon et al. (1996).*

*Fig.15. Pelagic (upper) and benthic (middle) pools of P (left) and N (right) in the inner (red), intermediate (orange), outer (turquoise) and entire (black) Stockholm Archipelago. The retention efficiencies (%) of the nutrient load from land and atmosphere are shown for the entire Stockholm Archipelago (lower), where the small peaks derive from leap years.*





## 9   Tables

*Table 1. Number of sampling occasions (Occ) during the number of years, number of months during each year, and number of depths levels that was frequently sampled at the different stations used for validation of model results. The position of the stations can be seen in Fig. 3.*

| ID | Station name | Basin name | Occ | Years* | Months | Depths** |
|----|--------------|------------|-----|--------|--------|----------|
| A | Nyvarp | Sandöfjärden | 209 | 23 | 8 | 14 |
| B | Kanholmsfjärden | Kanholmsfjärden | 206 | 23 | 9 | 13 |
| C | Solöfjärden | Solöfjärden | 213 | 23 | 8 | 14 |
| D | TrälhavetII | Trälhavet | 215 | 23 | 9 | 13 |
| E | S. Vaxholmsfjärden | S. Vaxholmsfjärden | 131 | 23 | 7 | 8 |
| F | Blomskär | Stora Värtan | 141 | 23 | 8 | 9 |
| G | Blockhusudden | Strömmen | 249 | 23 | 11 | 16 |
| H | Baggensfjärden | Baggensfjärden | 173 | 20 | 9 | 10 |

* Entire period is 23 years; **Sampled at least half of the sample occasions





*Table 2. The correlation coefficients (r) between observations (obs) and model results (S-HYPE), and the long term (1990-2012) averages of river outflow (QF) and nutrient loads from Lake Mälaren.*

| Variable | Units | Average obs | Average S-HYPE | r |
|----------|-------|-------------|----------------|-----|
| $Q_F$ | $10^6 \ m^3 \ month^{-1}$ | 421 | 422 | 0.94 |
| TN | $t \ month^{-1}$ | 270 | 271 | 0.93 |
| DIN | $t \ month^{-1}$ | 83 | 76 | 0.86 |
| TP | $t \ month^{-1}$ | 13 | 11 | 0.87 |
| DIP | $t \ month^{-1}$ | 5.7 | 5.7 | 0.79 |



*Table 3. The total land load (rivers, land run-off and atmosphere) of P and N (t yr$^{-1}$) to the Stockholm Archipelago, the size of the benthic and pelagic N and P pools (t), and the export from the area (t yr$^{-1}$) before and after the nutrient reductions, as well as their percentage changes. The system is in both cases in steady state, thus the benthic and pelagic pools are in balance with the nutrient load.*

|  |  | Unit | Initial values | End of period | Change (%) |
|---|---|---|---|---|---|
| **P** | Total load | t yr$^{-1}$ | 213 | 186 | -13 |
|  | Pool$^*$ | t | 10661 | 9952 | -7 |
|  | Export | t yr$^{-1}$ | -3.5 | -11 | -207 |
|  | Retention | % | 101 | 106 |  |
| **N** | Total load | t yr$^{-1}$ | 7690 | 6164 | -20 |
|  | Pool$^*$ | T | 35196 | 33216 | -6 |
|  | Export | t yr$^{-1}$ | 1585 | 609 | -62 |
|  | Retention | % | 79 | 90 |  |
| **N:P**$^{**}$ | Molar ratio |  | 42 | 35 | -16 |

*The sum of benthic and pelagic pools. ** In the inner archipelago.



*Table 4. The maximum concentrations of P and N (mg l$^{-1}$) in the discharge from sewage treatment plants of different size (person equivalents, pe) and the reduction of P and N loads from rivers and industries (%) in the scenario run with the SCM.*

| Sewage treatment facilities (pe) | P (mg l$^{-1}$) | N (mg l$^{-1}$) |
|---|---|---|
| >50 000 | 0.1 | 4 |
| 10 000-50 000 | 0.1 | 6 |
| <10 000 | 0.15 | 10 |
| **Decrease in land load** | (%) | (%) |
| Industries | 20 | 20 |
| River load | 20 | 29 |