# Peer review of "Modelling nutrient retention in the coastal zone of an"

_Biogeosciences, 2016_

## Referee Comment (RC1) · Anonymous Referee #1 · 25 Mar 2016

The manuscript Almroth-Rosell et al investigates nutrient retention in the eutrophic Stockholm Archipelago. A 1990-2012 simulation is first validated against observations and then used to calculate the retention of N and P in the three sub-basins of the archipelago. A sensitivity analysis with realistic N and P reduction is then carried out to assess the effect of nutrient management on nutrient retention. The study is regional but of general interest, the manuscript is well written and the method and overall analysis appropriate. I therefore recommend publication of the manuscript in Biogeosciences after minor to moderate revisions. My main comments are discussed below, followed by a list of specific comments.

My main issue is the lack of discussion of the results with respect to the wider literature. This is probably due to the choice of format that merge the Results with Discussion. A comparison of retention capacity with previous studies is provided (e.g. Figure 14) but

the authors should provide a deeper discussion of their results and their implications. This would benefit their study and the audience of Biogeosciences.

Nutrient load decrease during the study which lead to uncertainties in the interpretation of the results. Could the author calculate retention for specific periods (in addition to the 1990-2012 period), i.e. in the earlier and later period of the study, in order to remove the uncertainty about the system not being at steady state? or provide a retention time series for 1990-2012, as in Figure 15? This would also provide information about the change in the system over this period.

Specific comments:

P5L4: missing words, "which is why"?

P8L21: (2009; 2011)

P9L3-5: Not clear, does that mean that for each observation you calculate the most limiting nutrient (N or P) and infer the non limiting nutrient with respect to Redfield?

P9 L21: why is there a plus/minus symbol in front of the square root?

P12L3-5: Is this first sentence needed?

P12L6: suggestion: "good agreement" or "agrees well" rather than "good results"

P12L21: Please rephrase for clarity (and remove "described")

P13L3-5: Why choosing this station?

P13L8-10: Unless there is primary production in summer.

P13L22: "surface nutrients".

P13L25-29: Why not move this higher up, before you describe the first station. Also, change "parameters" to "state variables" (or else), also L31.

P15L4-5: This is already mentioned above.

P15L5-20: Discuss more the implications of the different estimates.

P15L31: suggestion: "during the simulated period" or "over the 1990-2012 period".

P15L31-P16L2. Could you be more specific about the importance of the import of nutrients? Also this statement sounds strange, you say that the effect of nutrient transport has not been studied and will be studied in the future. Then what is this mentioned at all?

P17L2-4: please rephrase for clarity.

P18L7: "by" instead of "with"

P18L29: Replace with "This implies that under the 2010 conditions, all the total P load..."

P20L5: Suggestion: change to "overall, model results agree with observations"

P20L15: Suggestion: start sentence with However.

P20L20: Space before N.

P20L27-28: Doesn't the retention capacity increases for N on short time scales?

---

## Referee Comment (RC2) · Anonymous Referee #2 · 29 Mar 2016

The model study of Almroth-Rosell et al. presents a sound model approach to improve our understanding of the nutrient retention in the archipelago oft he city of Stockholm. Through a combination of different models the authors estimate the retention capacities of nitrogen and phosphorous in different basins from nutrient sources to the Baltic Sea. The models which are combined here and the validation of the model is logical and well done, the results may be relevant for managers. The critical aspect here is the lack of a significant increase of our mechanistic understanding of processes leading to retention or the factors impacting retention. Moreover, the processes behind the retention are not clearly described and thus it remains difficult to fully understand how the retention works in this approach. The processes described are burial and for nitrogen also the process of microbial denitrification in sediments (probably also water column, when oxygen goes to depletion). Clarification of these aspects is required and some detailed

comments for improvement are given below.

The introduction needs more clarity and focus. The text jumps from general statements to specific Baltic Sea aspects e.g. hypoxia in the Gulf of Mexico is mentioned and next loads of nutrients to the Baltic Sea (p3 lines 8-14) or global eutrophication and loads to the Baltic Sea (p3 lines 18-22). I also did not understand why this example on retention (of nitrogen?) by plants is chosen (p4)? Why is the expression river outlet preferred over estuary (p4 line 18)? The description of the archipelago (p4 lines 23-27) is very brief and general. The more detailed description follows later. This should be combined or at least the reader should be referred to the later text. Further below the reduction of sewage is mentioned however again without details (line 31, reduction of how much N?). The a classification is mentioned (unsatisfactory eutrophic) but what is that really? What are high nutrient loads (p5 line 4)? On p.6 (lines 16-19) continue with somewhat vague site descriptions. Please add data on nutrient release over time and what the improvement of the treatment really means in concentration and load changes. Here again a combination of the text in the introduction with the study site description is would be better. What is the overall intention of this study – is it a retention estimate for managers, or are processes considered and their regulation by natural settings and anthropogenic impact? These are two different foci which impact the model set-up and the description of results. To me it seems that the authors mix both aspects with the results that neither aspect receives sufficient attention.

The model description is well done and clear in most cases. Minor requests for clarification are: The conversion of hydrogen sulfide into negative oxygen concentrations is sometimes used but seems not correct since it does not include the conversion of sulfate into hydrogen sulfide. What is the justification for this? How is the amount of N an P stored in sediments calculated (p 8 line 1)? How is nitrification and denitrification modelled (line 8)? Better focus the text lines 10-21 on the critical variables for this study, burial, remineralisation – assimilation and nitrogen fixation are less important. Is the atmospheric deposition indeed significant and deserves this much attention

throughout the text (see line 24)? Rivers hardly every supply nutrients with a N:P ratio of 16, usually the ratio is much higher (see p9 line 3-5). May be I misunderstand the calculation of the loads, but this needs conversion to true input ratios and concentration. The retention calculation (p11) is crucial for the manuscript. Please improve the description and explain R tot better, so that equation 4 becomes clearer. I could not understand the sentence p.12 line 22-23 about the validation of results and how representative stations are. Where are the cost functions from which are mentioned (p.13 line 25)? In case oxygen is not very well simulated by the model, then denitrification estimates cannot properly work in the model. How well is oxygen represented and how does that impact the results of the denitrification estimates? In this context the retention time of the water also plays a crucial role since longer retention times should lead to decreasing oxygen concentrations in the water. It would be good to dedicate a paragraph to the linkage of these variables and discuss the model results in relation to findings at the representative stations. In the present text the retention is qualitatively mentioned but quantifications are lacking (see paragraph 3.2.2). The paragraph on reduction scenarios is a pure description of model results but lacks aspects of a discussion – this would therefore need drastic shortening. It would be nice to significantly reduce number of figures.

Overall - as already mentioned above – the study would profit by a comparison of own results with other such model exercises. Although I understand why the paper was submitted to Biogeosciences it may be better placed in a model journal. As the paper stands now it does not explain the processes of nutrient retention or relate them to environmental processes (except the nutrient reduction scenario).

---

## Author Comment (AC1) · 20 May 2016

First we would like to thank the two reviewers for their helpful and thoughtful comments. In our response we repeat the comments/questions followed by our response. We refer to page + line numbers in the original manuscript if suited.

RC-Referee comment, AR-Author reply

Response to referee #1

RC: My main issue is the lack of discussion of the results with respect to the wider literature. This is probably due to the choice of format that merge the Results with Discussion. A comparison of retention capacity with previous studies is provided (e.g. Figure 14) but the authors should provide a deeper discussion of their results and their

implications. This would benefit their study and the audience of Biogeosciences.

AR: We agree with the referee and will in the revised manuscript deepen the discussion of the results and their implications. We will also with a wider perspective again search the scientific literature for similar model exercises to which our results can be compared.

RC: Nutrient load decrease during the study which lead to uncertainties in the interpretation of the results. Could the author calculate retention for specific periods (in addition to the 1990-2012 period), i.e. in the earlier and later period of the study, in order to remove the uncertainty about the system not being at steady state? or provide a retention time series for 1990-2012, as in Figure 15? This would also provide information about the change in the system over this period.

AR: Yes, we can include also a figure of the time series for 1990-2012. However, we think that there are enough numbers of figures, which is also commented by referee #2 why we will compromise and re-do figure nr 10. The new figure will show the total external nutrient load as well as the nutrient retention in the entire Stockholm Archipelago. The results which are not already discussed will be included in the discussion to improve the interpretation. The figure is uploaded to the discussion and here called Fig.1.

RC: Specific comments

AR: In the revised manuscript we will gratefully consider all the specific comments and accept the suggestions where appropriate.
* * *
**Fig. 1.**

---

## Author Comment (AC2) · 20 May 2016

First we would like to thank the two reviewers for their helpful and thoughtful comments. In our response we repeat the comments/questions followed by our response. We refer to page + line numbers in the original manuscript if suited.

RC-Referee comment, AR-Author reply.

Response to referee #2

RC: The model study of Almroth-Rosell et al. presents a sound model approach to improve our understanding of the nutrient retention in the archipelago of the city of Stockholm. Through a combination of different models the authors estimate the retention capacities of nitrogen and phosphorous in different basins from nutrient sources to

the Baltic Sea. The models which are combined here and the validation of the model is logical and well done, the results may be relevant for managers. The critical aspect here is the lack of a significant increase of our mechanistic understanding of processes leading to retention or the factors impacting retention. Moreover, the processes behind the retention are not clearly described and thus it remains difficult to fully understand how the retention works in this approach. The processes described are burial and for nitrogen also the process of microbial denitrification in sediments (probably also water column, when oxygen goes to depletion). Clarification of these aspects is required and some detailed comments for improvement are given below.

AC: We thank the referee for the detailed and positive review as well as for the suggested improvements. In the revised manuscript we will describe and discuss the different retention processes in more detail and how they are affected by e.g. oxygen depletion/reoxygenation.

RC: The introduction needs more clarity and focus. The text jumps from general statements to specific Baltic Sea aspects e.g. hypoxia in the Gulf of Mexico is mentioned and next loads of nutrients to the Baltic Sea (p3 lines 8-14) or global eutrophication and loads to the Baltic Sea (p3 lines 18-22).

AC: The aim of the introduction is to put the retention question and the eutrophication problem in a global perspective. To clarify, we will reorganize the introduction to first introduce the global perspective on eutrophication and coastal nutrient retention and thereafter describe the Baltic Sea and the model study in the Stockholm archipelago that is used as an example for more detailed discussions about the processes affecting nutrient retention.

RC: I also did not understand why this example on retention (of nitrogen?) by plants is chosen (p4)?

AC: To be able to calculate the retention in an area the definition of the word need to be discussed. Plants are assimilating nutrients, why it is removed from the water mass.

This removal is however not permanent but might lead to a higher degree of burial. We thank the reviewer for pointing out the shortcoming of the implication with this example and will clarify it in the revised manuscript.

RC: Why is the expression river outlet preferred over estuary (p4 line 18)?

AC: It was not our intention to choose one expression over another. In the revised manuscript we will change the expression to estuary.

RC: The description of the archipelago (p4 lines 23-27) is very brief and general. The more detailed description follows later. This should be combined or at least the reader should be referred to the later text.

AC: The Stockhom Archipelago is the name of the area. In the revised manuscript we will refer to the study site paragraph where it is described more in detail. We will also reorganize the text and move unnecessary details about the study site from the introduction to the methods.

RC: Further below the reduction of sewage is mentioned however again without details (line 31, reduction of how much N?).

AC: We agree that more details are needed. We will include a more extensive description in the revised manuscript and also reorganize the text and move unnecessary details about the study site from the introduction to the methods.

RC: The classification is mentioned (unsatisfactory eutrophic) but what is that really?

AC: We agree that a clarification of the expression is needed. We will change the formulation in the revised mansuscript and describe the environmental classification.

RC: What are high nutrient loads (p5 line 4)?

AC: We will rephrase the sentence in the revised manuscript and point out that the river Norrström supplies large nutrient loads to the inner part of the archipelago.

RC: On p.6 (lines 16-19) continue with somewhat vague site descriptions. Please add data on nutrient release over time and what the improvement of the treatment really means in concentration and load changes. Here again a combination of the text in the introduction with the study site description is would be better.

AC: We will reorganize the text to focus all the study site relevant descriptions in the methods section. We will move details from the introduction to the methods and also refer e.g. to figures of nutrients loads shown in the results section.

RC: What is the overall intention of this study – is it a retention estimate for managers, or are processes considered and their regulation by natural settings and anthropogenic impact? These are two different foci which impact the model set-up and the description of results. To me it seems that the authors mix both aspects with the results that neither aspect receives sufficient attention.

AC: The aim is to investigate the retention capacity of the coastal zone along the land to sea continuum, which is described in the aim-section in the end of the introduction. The aim is also to discuss the processes affecting the nutrient retention in the coastal areas as well as to discuss the concept of retention. We will have a closer look and rephrase/rewrite necessary parts to prevent the focus from being lost in the revised manuscript. We will therefore, as mentioned above, clarify more the processes leading to retention and the factors impacting retention.

RC: The model description is well done and clear in most cases. Minor requests for clarification are: The conversion of hydrogen sulfide into negative oxygen concentrations is sometimes used but seems not correct since it does not include the conversion of sulfate into hydrogen sulfide. What is the justification for this?

AC: We agree that this need some clarification, which will be done in the revised manuscript. The conversion of sulfate into hydrogen sulfide is included in the "negative oxygen". Sulfate is assumed to be reduced according to the model formulations but instead of stating the amount of hydrogen sulphide produced, the term "negative

oxygen" is used, corresponding to the amount of oxygen needed to oxidise the hydrogen sulphide.The cited reference in the manuscript will be changed to: Fonselius, S. H. (1969). Hydrography of the Baltic deep basins III.

RC: How is the amount of N an P stored in sediments calculated (p 8 line 1)?

AC: This will be better described in the revised manuscript. The sediment layer in the present model is described by one vertically integrated bulk sediment parameterization (level 3 in Soetart et al, 2000). The organic material sinking to the sediment is divided into one nitrogen pool and one phosphorus pool described by the state variables NBT and PBT, respectively. The sediment module includes burial and aggregated process descriptions for oxygen and temperature dependent nutrient regeneration and denitrification. Reference: Soetaert, K., Middelburg, J.J., Herman, P.M.J., Buis, K., 2000. On the coupling of benthic and pelagic biogeochemical models. Earth-Science Reviews 51, 173–201.

RC: How is nitrification and denitrification modelled (line 8)?

AC: Nitrification and denitrification are oxygen dependent processes that occur both in the pelagic and benthic parts of the model. The processes are described in detail in the paper by Eilola et al. 2009, (referred to in the manuscript). We agree that these processes may need a more extensive description since especially the denitrification is an important retention process. But we will in this paper not repeat details about processes descriptions. Instead we want to focus on the relative impact of different processes on the nutrient retention and try to describe that even better as mentioned above.

RC: Better focus the text lines 10-21 on the critical variables for this study, burial, remineralisation – assimilation and nitrogen fixation are less important.

AC: This is a brief general description how the model works to help the reader, if not used to models, to understand that these processes are included. The paragraph does

not aim to describe the processes in details.

RC: Is the atmospheric deposition indeed significant and deserves this much attention throughout the text (see line 24)?

AC: We believe that the reader should be informed about the forcing that drives the model. Understanding the relative importance of the different drivers we consider important as well. Atmospheric deposition is one even though it is not a dominant source of nutrients in the inner archipelago. However, in the outer archipelago, which has a large area, the atmospheric input of nutrients actually is the dominant external source.

RC: Rivers hardly every supply nutrients with a N:P ratio of 16, usually the ratio is much higher (see p9 line 3-5). May be I misunderstand the calculation of the loads, but this needs conversion to true input ratios and concentration.

AC: Thank you for pointing out this shortcoming of the description. The external inorganic nutrient loads are added as DIN and DIP from the forcing as they are in the model. This will be better described in the revised manuscript. The bioavailability and the composition (dissolved or particulate) of the organic nitrogen and phosphorus loading from land are generally not known. In the present model configuration the part of organic nutrient loads following the Redfield ratio are assumed to be bioavailable. The nutrient supply that fulfils the Redfield ratio is therefore added to the detritus pool in the model while the remaining fractions of nutrient loads are treated as conservative tracers in the model. This will be better explained in the revised manuscript.

RC: The retention calculation (p11) is crucial for the manuscript. Please improve the description and explain R tot better, so that equation 4 becomes clearer.

AC: Yes, thank you, we agree that this is a very important paragraph for the understanding of the study, and will improve the description in the revised manuscript.

RC: I could not understand the sentence p.12 line 22-23 about the validation of results and how representative stations are.

AC: We agree that this sentence could be re-formulated for better understanding, which we will do in the revised manuscript. What we intend to describe is that the local conditions of stations where data are observed are very important as they can vary due to the positions of the stations. When analyzing model results and interpreting validation results it is important to be aware that the model integrates the over a large area, while observations are sampled only at one position that possibly is not representative for the average conditions.

RC: Where are the cost functions from which are mentioned (p.13 line 25)?

AC: Thank you for the remark; we will include a cross-reference to equation three in the revised manuscript.

RC: In case oxygen is not very well simulated by the model, then denitrification estimates cannot properly work in the model. How well is oxygen represented and how does that impact the results of the denitrification estimates?

AC: Thank you for the comment. The capacity of the model to simulate oxygen is acceptable at all stations except for basin "Sandöfjärden" where the vertical summer profile shows not good results. This can be seen in figure 8 (circles). However, the impact of different oxygen conditions should of course be discussed more in detail in the discussion section.

RC: In this context the retention time of the water also plays a crucial role since longer retention times should lead to decreasing oxygen concentrations in the water. It would be good to dedicate a paragraph to the linkage of these variables and discuss the model results in relation to findings at the representative stations.

AC: Thank you for the comment! Yes, we will include a discussion of this in the revised manuscript. The largest sensitivity of denitrification rates to changes in oxygen concentrations occur at concentrations below 4-5 ml/l in the model. The maximum denitrification rate is obtained at an oxygen concentration of about 1 ml/l while denitrification

halts under anoxic conditions. We will investigate in which areas of the archipelago the model denitrification may be most sensitive to changes of the oxygen concentrations.

RC: In the present text the retention is qualitatively mentioned but quantifications are lacking (see paragraph 3.2.2).

AC: Quantification of the modelled retention are described and discussed in section 3.2.1. Quantifications of retention in other studies are shown in fig. 14 in section 3.2.2. In the revised manuscript we will in more detail discuss the different types of coastal areas and their retention capacities. We will quantify how water depth and residence time and mean bottom water oxygen concentrations affect the modelled nutrient retention.

RC: The paragraph on reduction scenarios is a pure description of model results but lacks aspects of a discussion – this would therefore need drastic shortening.

AC: We agree that the description of results are somewhat more extensive compared to the implications of the same. In the revised manuscript we will improve the discussion and better describe the implications of the reduction scenario experiment. The description of the setup of scenarios will be moved to methods and the discussions will be focused and more related also to the discussions about processes affecting retention.

RC: It would be nice to significantly reduce number of figures.

AC: Yes, we are aware that the number of figures is high. However, referee nr 1 actually wanted additional one figure. We will compromise and re-do figure nr 10. The new figure will show the total external nutrient load in the entire Stockholm archipelago and in addition the nutrient retention in the entire Stockholm Archipelago. The results which are not already discussed will be included in the discussion to improve the interpretation. The figure is separately uploaded to the discussion. We will also remove fig. nr 13, since the results are shown also in fig. 14. The text will be adjusted to fit the changes.

RC: Overall - as already mentioned above – the study would profit by a comparison of own results with other such model exercises.

AC: We have searched the scientific literature and have so far not found any similar model studies for the coastal zone. However, we will do a wider search for model exercises of nutrient budgets and retention from the literature and relate their results to our study.

RC: Although I understand why the paper was submitted to Biogeosciences it may be better placed in a model journal. As the paper stands now it does not explain the processes of nutrient retention or relate them to environmental processes (except the nutrient reduction scenario).

AC: In the revised manuscript we will overall improve the discussion of the nutrient retention processes and the impact from e.g. oxygen changes. We also think that the study has it main interests in the coastal zone as a filter, and the retention processes and not only as a modelling paper, even though we use a model as a tool. For instance, the importance of permanent relative to temporal retention need a clarification of the processes involved that can only be quantified and explained from dynamic model results including the pools of nutrients both in the water and in the sediment as in the present study. Also the impact from changing environmental conditions such as anticipated nutrient load reductions may best be explained from model experiments. These results are already quite substantial but we embrace the criticism from the reviewer and will according to the discussions above put more emphasis on trying to explain the environmental factors that impact on the processes causing nutrient retention. We will also better discuss how our findings relate to other studies found in the literature.

Additional planned changes:

AC: We have also noticed that we in the figures use different units, which will be changed in the revised manuscript.

---

## Author Response (AR1)

**This file contains:**

1. **A point-by-point response to the referees, including a list of relevant changes**
2. **A marked up manuscript version, thus, the manuscript including trackchanges.**

**Response o referees**

2016-06-30

First we would like to thank the two reviewers for their helpful and thoughtful comments. In our response we repeat the comments/questions followed by our response/changes. We refer to page + line numbers in the revised manuscript in which "track changes" are included, if suited.

RC-Referee comment

AR-Author reply

**Response to referee #1**

**RC:** My main issue is the lack of discussion of the results with respect to the wider literature. This is probably due to the choice of format that merge the Results with Discussion. A comparison of retention capacity with previous studies is provided (e.g. Figure 14) but the authors should provide a deeper discussion of their results and their implications. This would benefit their study and the audience of Biogeosciences.

**AR:** We have deepened the discussion of the results and their implications in the revised manuscript. We have also, with a wider perspective, again searched the scientific literature for similar model exercises to which our results have been compared.

**RC:** Nutrient load decrease during the study which lead to uncertainties in the interpretation of the results. Could the author calculate retention for specific periods (in addition to the 1990-2012 period), i.e. in the earlier and later period of the study, in order to remove the uncertainty about the system not being at steady state? or provide a retention time series for 1990-2012, as in Figure 15? This would also provide information about the change in the system over this period.

**AR:** Retention time series for 1990-2012 are now included. Since we think that there are enough numbers of figures, which is also commented by referee #2 we have compromised and re-made figure nr 10. The new figure shows the total external nutrient load as well as the nutrient retention in the entire Stockholm Archipelago.

The results which are not already discussed will be included in the discussion to improve the interpretation, P14 L1-3.

**RC:** Specific comments

**RC:** P5L4: missing words, "which is why"?

**AR:** This section is re-organized and sentence above is removed.

**RC:** P8L21: (2009; 2011)

**AR:** This is changed. P6L10

**RC:** P9L3-5: Not clear, does that mean that for each observation you calculate the most limiting nutrient (N or P) and infer the non limiting nutrient with respect to Redfield?

**AR:** This section is now rewritten, see P8L22-29 in the revised manuscript. The external inorganic nutrient loads are added as DIN and DIP from the forcing as they are in the model. The bioavailability and the composition (dissolved or particulate) of the organic nitrogen and phosphorus loading from land are generally not known. In the present model configuration the part of organic nutrient loads following the Redfield ratio are assumed to be bioavailable. The nutrient supply that fulfils the Redfield ratio is therefore added to the detritus pool in the model while the remaining fractions of nutrient loads are treated as conservative tracers in the model.

**RC:** P9 L21: why is there a plus/minus symbol in front of the square root?

**AR:** We have removed the plus/minus symbol, it is not needed in Eq. 2 in the revised manuscript, P9L13

**RC:** P12L3-5: Is this first sentence needed?

**AR:** The sentence is removed in the revised manuscript.

**RC:** P12L6: suggestion: "good agreement" or "agrees well" rather than "good results"

**AR:** The suggestion is accepted and the sentence is re-written. See P13L3-4 in the revised manuscript.

**RC:** P12L21: Please rephrase for clarity (and remove "described")

**AR:** The paragraph is now re-written, see P13L17-21 in the revised manuscript. What we intend to describe is that the local conditions of stations where data are observed are very important as they can vary due to the positions of the stations. When analyzing model results and interpreting validation results it is important to be aware that the model integrates over a large area, while observations are sampled only at one position that possibly is not representative for the average conditions.

**RC:** P13L3-5: Why choosing this station?

**AR:** This station was chosen because the observation data set is largest at this station. This is now added in the revised manuscript, P13L28-30.

**RC:** P13L8-10: Unless there is primary production in summer.

**AR:** We did a check of the observation data during summer and there are no significant over saturation of oxygen due to primary production. A sentence discussing the reason is now added since there were only measurements from November (with quite high tempertures) and February. See P14L13-17 in the revised manuscript.

**RC:** P13L22: "surface nutrients".

**AR:** The word "surface" is added. See P14L30 in the revised manuscript.

**RC:** P13L25-29: Why not move this higher up, before you describe the first station. Also, change "parameters" to "state variables" (or else), also L31.

**AR:** This paragraph is now moved upwards in the revised manuscript and "parameters" is changed to "state variables" in all places, P14L1-6.

**RC:** P15L4-5: This is already mentioned above.

**AR:** This is now removed in the revised manuscript.

**RC:** P15L5-20: Discuss more the implications of the different estimates.

**AR:** An extended discussion is now included in the revised manuscript, P16.

**RC:** P15L31: suggestion: "during the simulated period" or "over the 1990-2012 period".

**AR:** The word "simulated" is added. See P17L11 in the revised manuscript.

**RC:** P15L31-P16L2. Could you be more specific about the importance of the import of nutrients?

**AR:** This paragraph is now changed in the revised manuscript, P17L11-16.

**RC:** Also this statement sounds strange, you say that the effect of nutrient transport has not been studied and will be studied in the future. Then what is this mentioned at all?

**AR:** We agree and have removed this sentence.

**RC:** P17L2-4: please rephrase for clarity.

**AR:** This sentence is not needed and a description on what residence time and how it was calculated is moved to the method section 2.4. See P12L4-7 in the revised manuscript.

**RC:** P18L7: "by" instead of "with"

**AR:** Done. This section, describing the reduction scenario was also moved to section 2.4.1. See P12L9-26 in the revised manuscript.

**RC:** P18L29: Replace with "This implies that under the 2010 conditions, all the total P load..."

**AR:** Done. See P20L29-30 in the revised manuscript.

**RC:** P20L5: Suggestion: change to "overall, model results agree with observations"

**AR:** Done. See P23L4-5 in the revised manuscript.

**RC:** P20L15: Suggestion: start sentence with However.

**AR:** The sentence starts now with "Thus". See P23L15 in the revised manuscript.

**RC:** P20L20: Space before N.

**AR:** Done.

**RC:** P20L27-28: Doesn't the retention capacity increases for N on short time scales?

**AR:** Yes, but not for P. This is now clarified in the revised manuscript, P23L27-28.

**Response to referee #2**

**RC:** The model study of Almroth-Rosell et al. presents a sound model approach to improve our understanding of the nutrient retention in the archipelago of the city of Stockholm. Through a combination of different models the authors estimate the retention capacities of nitrogen and phosphorous in different basins from nutrient sources to the Baltic Sea. The models which are combined here and the validation of the model is logical and well done, the results may be relevant for managers. The critical aspect here is the lack of a significant increase of our mechanistic understanding of processes leading to retention or the factors impacting retention. Moreover, the processes behind the retention are not clearly described and thus it remains difficult to fully understand how the retention works in this approach. The processes described are burial and for nitrogen also the process of microbial denitrification in sediments (probably also water column, when oxygen goes to depletion). Clarification of these aspects is required and some detailed comments for improvement are given below.

**AC:** We thank the referee for the detailed and positive review as well as for the suggested improvements. In the revised manuscript we describe and discuss the different retention processes in more detail and how they are affected by e.g. oxygen depletion/reoxygenation.

**RC:** The introduction needs more clarity and focus. The text jumps from general statements to specific Baltic Sea aspects e.g. hypoxia in the Gulf of Mexico is mentioned and next loads of nutrients to the Baltic Sea (p3 lines 8-14) or global eutrophication and loads to the Baltic Sea (p3 lines 18-22).

**AC:** The aim of the introduction is to put the retention question and the eutrophication problem in a global perspective. We have reorganized the introduction to first introduce the global perspective on eutrophication and coastal nutrient retention with both Gulf of Mexico and the Baltic Sea as examples in some cases. Thereafter we describe the Baltic Sea and the model study in the Stockholm archipelago that is used as an example for more detailed discussions about the processes affecting nutrient retention. Specific information about the Stockholm Archipelago is moved to the Method chapter (chapter 2), in the study site section (section 2.1).

**RC:** I also did not understand why this example on retention (of nitrogen?) by plants is chosen (p4)?

**AC:** To be able to calculate the retention in an area the definition of the word need to be discussed. Plants are assimilating nutrients, leading to a removal of the dissolved nutrients from the water mass, which however is not permanent. The transformation of dissolved nutrients into organic material might lead to another exposure of the nutrients for other biogeochemical processes e.g. higher degree of burial. This is clarified in the revised manuscript, P3L25-27, and also more extensively discussed in section 3.2.2.

**RC:** Why is the expression river outlet preferred over estuary (p4 line 18)?

**AC:** It was not our intention to choose one expression over another. We have changed to "estuary" in the revised manuscript, P4L11.

**RC:** The description of the archipelago (p4 lines 23-27) is very brief and general. The more detailed description follows later. This should be combined or at least the reader should be referred to the later text.

**AC:** We have re-organized the text and moved unnecessary details about the study site from the introduction to the methods in the revised manuscript. In the introduction we now refer to the study site paragraph (section 2.1) for more details about the Stockholm Archipelago.

**RC:** Further below the reduction of sewage is mentioned however again without details (line 31, reduction of how much N?).

**AC:** We have included a more extensive description in the revised manuscript and also reorganized the text and moved unnecessary details about the study site from the introduction to the methods (section 2.1).

**RC:** The classification is mentioned (unsatisfactory eutrophic) but what is that really?

**AC:** We agree that a clarification of the expression was needed. This section is now extended to include short information about the classification in the revised manuscript, P6L1-7.

**RC:** What are high nutrient loads (p5 line 4)?

**AC:** After a reorganization of the introduction parts of the text is moved to the study site section. This sentence is not needed anymore and was therefore removed.

**RC:** On p.6 (lines 16-19) continue with somewhat vague site descriptions. Please add data on nutrient release over time and what the improvement of the treatment really means in concentration and load changes. Here again a combination of the text in the introduction with the study site description is would be better.

**AC:** We have included a more extensive description on the nutrient reductions in the revised manuscript, section 2.1.

**RC:** What is the overall intention of this study – is it a retention estimate for managers, or are processes considered and their regulation by natural settings and anthropogenic impact? These are two different foci which impact the model set-up and the description of results. To me it seems that the authors mix both aspects with the results that neither aspect receives sufficient attention.

**AC:** The aim is to investigate the filter efficiency of the coastal zone along the land to sea continuum, which is described in the aim-paragraph in the end of the introduction. The aim is also to discuss the processes affecting the nutrient retention in the coastal areas as well as to discuss the concept of retention. We have rephrased and rewrote necessary parts to prevent the focus from being lost in the revised manuscript.

**RC:** The model description is well done and clear in most cases. Minor requests for clarification are: The conversion of hydrogen sulfide into negative oxygen concentrations is sometimes used but seems not correct since it does not include the conversion of sulfate into hydrogen sulfide. What is the justification for this?

**AC:** We agree that this needed some clarification, which is done in the revised manuscript, P7L17-19. The conversion of sulfate into hydrogen sulfide is included in the "negative oxygen". Sulfate is assumed to be reduced according to the model formulations but instead of stating the amount of hydrogen sulphide produced, the term "negative oxygen" is used, corresponding to the amount of oxygen needed to oxidise the hydrogen sulphide.The cited reference in the manuscript is changed to: Fonselius, S. H.: Hydrography of the Baltic deep basins III, 1969.

**RC:** How is the amount of N and P stored in sediments calculated (p 8 line 1)?

**AC:** This is better described in the revised manuscript, P7L19-22. The sediment layer in the present model is described by one vertically integrated bulk sediment parameterization (level 3 in Soetart et al, 2000). The organic material sinking to the sediment is divided into one nitrogen pool and one phosphorus pool described by the state variables NBT and PBT, respectively. The sediment module includes burial and aggregated process descriptions for oxygen and temperature dependent nutrient regeneration and denitrification.
Reference: Soetaert, K., Middelburg, J.J., Herman, P.M.J., Buis, K., 2000. On the coupling of benthic and pelagic biogeochemical models. Earth-Science Reviews 51, 173–201.

**RC:** How is nitrification and denitrification modelled (line 8)?

**AC:** Nitrification and denitrification are oxygen dependent processes that occur both in the pelagic and benthic parts of the model. The processes are described in detail in the paper by Eilola et al. 2009, (referred to in the manuscript). We agree that these processes may need a more extensive description since especially the denitrification is an important retention process. But we will in this paper not repeat details about processes descriptions. Instead we focus on the relative impact of different processes on the nutrient retention and which is now more discussed in section 3.2.2.

**RC:** Better focus the text lines 10-21 on the critical variables for this study, burial, remineralisation – assimilation and nitrogen fixation are less important.

**AC:** This is a brief general description how the model works to help the reader, if not used to models, to understand that these processes are included. The paragraph does not aim to describe the processes in details.

**RC:** Is the atmospheric deposition indeed significant and deserves this much attention throughout the text (see line 24)?

**AC:** We believe that the reader should be informed about the forcing that drives the model. However, in the revised manuscript we have changed the order of the different forcing variables so that the atmospheric deposition of nutrients is the terminative variable in the sentence, P8L13-15. Understanding the relative importance of the different drivers we consider as important as well. Atmospheric deposition is one even though it is not a dominant source of nutrients in the inner archipelago. However, in the outer archipelago, which has a large area, the atmospheric input of nutrients actually is the dominant external source.

**RC:** Rivers hardly every supply nutrients with a N:P ratio of 16, usually the ratio is much higher (see p9 line 3-5). May be I misunderstand the calculation of the loads, but this needs conversion to true input ratios and concentration.

**AC:** Thank you for pointing out this shortcoming of the description. The external inorganic nutrient loads are added as DIN and DIP from the forcing as they are in the model. This is better described in the revised manuscript, P8L22-29. The bioavailability and the composition (dissolved or particulate) of the organic nitrogen and phosphorus loading from land are generally not known. In the present model configuration the part of organic nutrient loads following the Redfield ratio are assumed to be bioavailable. The nutrient supply that fulfils the Redfield ratio is therefore added to the detritus pool in the model while the remaining fractions of nutrient loads are treated as conservative tracers in the model.

**RC:** The retention calculation (p11) is crucial for the manuscript. Please improve the description and explain R tot better, so that equation 4 becomes clearer.

**AC:** We agree that this is a very important paragraph for the understanding of the study, and have improved the description in the revised manuscript, section 2.4. Also, a supplement 2 is created where different retention calculations are defined.

**RC:** I could not understand the sentence p.12 line 22-23 about the validation of results and how representative stations are.

**AC:** The paragraph is now re-written; see P13L17-21 in the revised manuscript. What we intend to describe is that the local conditions of stations where data are observed are very important as they can vary due to the positions of the stations. When analyzing model results and interpreting validation results it is important to be aware that the model integrates over a large area, while observations are sampled only at one position that possibly is not representative for the average conditions.

**RC:** Where are the cost functions from which are mentioned (p.13 line 25)?

**AC:** Thank you for the remark. We have included cross-reference to Eq. 2 for the correlation coefficient and to Eq. 3 for the cost function value. The paragraph is also moved upwards in the text in the revised manuscript, P14L1-6, which also led to a re-organization of figures 6-8.

**RC:** In case oxygen is not very well simulated by the model, then denitrification estimates cannot properly work in the model. How well is oxygen represented and how does that impact the results of the denitrification estimates?

**AC:** Thank you for the comment. The capacity of the model to simulate oxygen is acceptable at all stations except for basin "Sandöfjärden" where the vertical summer profile shows not good results. This can be seen in figure 6 (circles) in the revised manuscript. However, the impact of changed oxygen conditions is discussed more in detail in the section 3.2.2.

**RC:** In this context the retention time of the water also plays a crucial role since longer retention times should lead to decreasing oxygen concentrations in the water. It would be good to dedicate a paragraph to the linkage of these variables and discuss the model results in relation to findings at the representative stations.

**AC:** Thank you for the comment! We have included a discussion of this in the revised manuscript, section 3.2.2. The largest sensitivity of denitrification rates to changes in oxygen concentrations occur at concentrations below 4-5 ml/l in the model. The maximum denitrification rate is obtained at an oxygen concentration of about 1 ml/l while denitrification halts under anoxic conditions. We also made a model experiment in which we investigated in which areas of the archipelago the model denitrification may be most sensitive to changes of the oxygen concentrations.

**RC:** In the present text the retention is qualitatively mentioned but quantifications are lacking (see paragraph 3.2.2).

**AC:** Quantification of the modelled retention are described and discussed in section 3.2.1. Quantifications of retention in other studies are shown in fig. 14 in section 3.2.2. In the revised manuscript we discuss in more detail the different types of coastal areas and their retention capacities. We have also added quantification on how water depth and residence time and mean bottom water oxygen concentrations affect the modelled nutrient retention.

**RC:** The paragraph on reduction scenarios is a pure description of model results but lacks aspects of a discussion – this would therefore need drastic shortening.

**AC:** The description of the setup of scenarios is moved to the methods chapter (section 2.4.2) in the revised manuscript. The discussion is in the revised manuscript more focused and more related also to the discussions about processes affecting retention.

**RC:** It would be nice to significantly reduce number of figures.

**AC:** Yes, we are aware that the number of figures is high. However, referee nr 1 actually wanted additional one figure. We compromised and re-made figure nr 10. The new figure shows the total external nutrient load in the entire Stockholm archipelago and in addition the nutrient retention in the entire Stockholm Archipelago. We also removed fig. nr 13 in the original manuscript, since the results are shown also in fig. 14. The text and the figure numbers are adjusted to fit the changes in the revised manuscript.

**RC:** Overall - as already mentioned above – the study would profit by a comparison of own results with other such model exercises.

**AC:** We have searched the scientific literature and have so far only found a few model studies for the coastal zone, there are however many model studies on lakes and catchment areas. There are a few empirical retention calculations which we discuss and e.g. Nielsen et al (2001) performed a retention study using a 2-layer hydrodynamic channel model in Randers Fjord, Denmark.

**RC:** Although I understand why the paper was submitted to Biogeosciences it may be better placed in a model journal. As the paper stands now it does not explain the processes of nutrient retention or relate them to environmental processes (except the nutrient reduction scenario).

**AC:** In the revised manuscript we have overall improved the discussion of the nutrient retention processes and the impact from e.g. oxygen changes. We also think that the study has it main interests in the coastal zone as a filter, and the retention processes and not only as a modelling paper, even though we use a model as a tool.

For instance, the importance of permanent relative to temporal retention need a clarification of the processes involved that can only be quantified and explained from dynamic model results including the pools of nutrients both in the water and in the sediment as in the present study. Also the impact from changing environmental conditions such as anticipated nutrient load reductions may best be explained from model experiments. These results are already quite substantial but we embrace the criticism from the reviewer and have put more emphasis on trying to explain the environmental factors that impact on the processes causing nutrient retention. We have also tried to better discuss how our findings relate to other studies found in the literature.

**Additional changes:**

We have also noticed that we in the figures used different units, which has been changed in the revised manuscript.

A definition of the retention parameters and way of calculating them are included as a supplement and the text is changed according to these definitions where suited.

[revised manuscript text omitted]

*The sum of benthic and pelagic pools. ** In the inner archipelago.

Borttaget: 3
Borttaget: and
Formaterat: Nedsänkt
Borttaget: Retention
Formaterat: Nedsänkt
Borttaget: Retention

**Borttaget:** ¶
¶
*Table 4. The maximum concentrations of P and N (mg l⁻¹) in the discharge from sewage treatment plants of different size (person equivalents, pe) and the reduction of P and N loads from rivers and industries (%) in the scenario run with the SCM.* ¶
**Sewage treatment facilities (pe)**  …

---

## Author Response (AR2)

**This file contains:**

1. **A point-by-point response to the editor and the referee, including a list of relevant changes**
2. **A marked up manuscript version, thus, the manuscript including "track changes" from the present revision.**

**Response to editor and referee**

2016-09-10

First we would like to thank the editor Prof. Fennel and the anonymous referee for their helpful and thoughtful comments. In our response we repeat the comments/questions followed by our response/changes. We refer to page + line numbers in the now revised manuscript, if suited, in which "track changes" are included.

AR-Author reply

**Editors comments:**

**Associate Editor Decision: Reconsider after major revisions** (30 Jul 2016) by Prof. Katja Fennel
Comments to the Author:
Dear Authors,

Reviewer 1 provided a further detailed review of your revised manuscript. I encourage you to carefully consider these comments.
In addition I'm providing my own comments below. I encourage you to submit a revised manuscript for further consideration as well as detailed responses to each comment.

Page and line numbers refer to the revised manuscript with changes tracked.

Define what you mean by "coastal filter" (P1, L18; P1, L25 and L26). The definition provided on page 4, lines 6-8 is insufficient. Where do these nutrients ultimately go? Which processes are responsible?
AR: This is now shortly better defined already in the abstract (P1, L19-21; P2, L2-5), with a more thorough definition in the introduction (P3, L17-21). It is clarified that retention in this manuscript also includes removal of nutrients (P3-4, L27-5), since the point of view is how much of the nutrient load from land does not reach the open sea.

P1, L24: "good quality" Do you mean "the simulated values agree well with observations"? Unclear otherwise. Please rephrase.
AR: Yes, this is what we mean and it is now rephrased (P1,L25-27).

P1, L26: "take care of" is not a good word choice. Replace by "remove" or "retain" or similar.
AR: We agree and have changed it (P2, L1).

Also, please explain where all the nutrients ultimately go. The Archipelago cannot infinitely continue to retain nutrients.
AR: This is now more clear in the abstract (P2, L2-5) and better described in the introduction (mentioned above (P3-4, L27-5)). Fig. 4 in the previous manuscript schematically shows the coastal filter and the retention and is therefore now moved to the introduction section and re-numbered to Fig. 1. Consequently the following figures up to fig. 3 changes number as well.

P2, L1-7: Denitrification does not "retain" nutrients. Denitrification removes nutrients from the system! This is different from retention.

AR: We agree with the Prof. Fennel that it is a simplification to include the denitrification process in the retention definition. But, since one of the "simpler" methods to calculate retention is to calculate the difference between the load of nutrients and the outflow (described in section 2.4), which is used in a number different studies, the comparable numbers of the retention includes denitrification. From the point of view that the filter capacity is defined as the difference from what comes in to the area and what comes out from the area, then the denitrification process is included in the retention. We also consider e.g. burial to be a removal of nutrients since it is a sink for the model. This is now better explained in the manuscript where the definition and explanation is described in the introduction and further described also in the method section (section 2.4).

P4, L8: "There are not enough estimates of nutrient retention in different coastal systems around the world…"
I note that you are not providing references of those that exist – something that was also criticized by Reviewer 1. Here are just three examples that I'm familiar with (I'm sure there are many more):

Xue, Z., He, R., Fennel, K., Cai, W.-J., Lohrenz, S., and Hopkinson, C., Modeling ocean circulation and biogeochemical variability in the Gulf of Mexico, Biogeosciences, 10, 7219-7234, doi:10.5194/bg-10-7219-2013 (2013)

Fennel, K., The role of continental shelves in nitrogen and carbon cycling: Northwestern North Atlantic case study. Ocean Science 6, 539-548, doi:10.5194/os-6-539-2010 (2010)

Fennel, K., Wilkin, J., Levin, J., Moisan, J., O'Reilly, J., Haidvogel, D., Nitrogen cycling in the Mid Atlantic Bight and implications for the North Atlantic nitrogen budget: Results from a three-dimensional model. Global Biogeochemical Cycles 20, GB3007, doi:10.1029/2005GB002456. (2006)

Two other studies of some potential relevance as well as the references given within are:

Laurent, A., Fennel, K., Hu, J., and Hetland, R.: Simulating the effects of phosphorus limitation in the Mississippi and Atchafalaya River plumes, Biogeosciences, 9, 4707-4723, doi:10.5194/bg-9-4707-2012 (2012)

Laurent, A., Fennel, K., Simulated reduction of hypoxia in the northern Gulf of Mexico due to phosphorus limitation, Elementa 2:000022, doi:10.12952/journal.elementa.000022 (2014)

Please make an effort to properly place your findings in the context of the published literature.
AR: First we would like to thank Prof. Fennel for the literature suggestions. We have read them all and considered their interesting results. We have included results from some of the studies where both the river input and the removal of nutrients (retention) are given, section 3.2 (P18, L25-31) as well as given examples of model studies in the introduction (P4, L24-25). We have also changed the formulation of the statement (P4, L17-18). New references included in the revised manuscript:

Fennel, K., Wilkin, J., Levin, J., Moisan, J., O'Reilly, J., Haidvogel, D., Nitrogen cycling in the Mid Atlantic Bight and implications for the North Atlantic nitrogen budget: Results from a three-dimensional model. Global Biogeochemical Cycles 20, GB3007, doi:10.1029/2005GB002456. (2006)

Seitzinger, S. P. and Giblin, A. E.: Estimating denitrification in North Atlantic continental shelf sediments, Biogeochemistry, 35, 235-260, 1996.

Xue, Z., He, R., Fennel, K., Cai, W.-J., Lohrenz, S., and Hopkinson, C., Modeling ocean circulation and biogeochemical variability in the Gulf of Mexico, Biogeosciences, 10, 7219-7234, doi:10.5194/bg-10-7219-2013 (2013)

P11, L13: Please see my earlier comment about the difference between removal and retention. Please be very clear what you mean. At present these two terms are lumped together which is not appropriate. Always be clear about the processes that are underlying your results. Just quantifying the difference between incoming and outgoing matter fluxes without explaining where the missing matter ended up is not sufficient.

AR: We do not totally agree. The retention has in many other studies been calculated as the difference between the nutrient load and the outflow e.g. Karlson (2010), Johnston (1991), Billen et al. (2011), and some other studies that we refer to in the manuscript. In a lot of areas the different processes are not known and a simpler method is the only choice. In our study we use both the simpler method, but we also calculate separately the burial and the denitrification processes. In this way we can compare the total "retention" with other studies. Also in Nixon et al. (1996), see e.g. fig. 3, the difference between the input and the outflow is defined as "retention", and a part of that retained nitrogen is denitrified. From the point of view that fractions of the nutrients supply from land are prevented from being further transported to the open sea, then it is "retained" in the area and for nitrogen that includes both burial and denitrification in our model study. As mentioned above, this is now more described in the introduction (P3-4, L25-13) as well as in the section 2.4 (P11, 11-27).

P11, L16: Do not relegate important definitions to the supplement. Define retention in the main text.

AR: Yes, we agree. Because the used retention definitions are already described in the main text the supplementary 2 is not necessary to include. We remove it from the submission.

P12: I question your calculation of "freshwater" residence time. As per the Monsen reference you provide, residence time is defined as the ratio of an inventory and it's inflow or outflow rate. You talk about freshwater residence time and divide the volume by the freshwater inflow rate, but the volume is not the freshwater volume but the total volume. Perhaps you do not mean freshwater residence time? Please clarify.

AR: We agree. We do mean the fresh water residence time and have now calculated is as the fresh water volume (given by a freshwater tracer in the model) dived by the freshwater inflow ($Q_f$). The description is revised in section 2.4 (P12, L14-23). This change also led to an updated Fig. 13 since the lower residence time affected the positions of our data points in the graph as well as the text in section 3.2.2 (P19-20, L25-1).

P12, L9-22: This text and Table 2 are not very informative. It is not clear what the baseline load is. Without this it is not particularly useful to state a percentage reduction. Consider giving less detail about treatment plants and instead describe better what nutrient inputs were used in the different experiments.

AR: We agree. The baseline load (forcing from 2010, the annual loads of N and P) are now given in the revised text in section 2.4.2 (P13). The level of details has been reduced. Also table 2 has been reduced since some of the numbers are now given in the text.

P13, L4-6: "A clear relationship between river outflow and nutrient load is observed …"
This is not a result that can be observed, but something that is imposed!!! Does not belong in the results but in the Methods.

AR: This sentence is now re-formulated (P14, L4-6). We hope the re-formulation makes this clearer.

Fig. 9: Please separate input from land and atmosphere instead of providing one combined number.

AR: Done.

The concept of temporary retention does not make much sense to me. Do you consider your system to be in a dynamic steady state? How do you distinguish between processes leading to temporary versus permanent retention? Do the N and P inventories in the water column differ between the beginning and end of the simulation period? Please clarify this in section 3.2.

AR: Permanent retention for P means burial. For N benthic denitrification represented as much as almost 92 % of the permanent retention, burial for less than 8 % and pelagic denitrification was below 1 %. This is written on P17, L7-10.

The temporary retention on the other hand is considered to be changes in the pelagic and benthic pools of nutrients. The nutrient pools include both the inorganic and organic nutrients. Factors that affect the benthic N and P pool are supply of organic material from the water column settling on the sediment, the decomposition of organic material and the release of inorganic nutrients as well as by burial (which in the model is a permanent sink of nutrients). The pelagic N and P pools are affected by the supply from land, the sinking of organic material, the release of nutrients from the sediment to the water column and to the net export of nutrients to areas down-stream. These processes are handled by the model and what we present in the paper is the total temporary retention. This is now written in section 2.4 (P11, L20-27).

The benthic and the pelagic pools can thus change during the simulation period, which is described in section 3.2 (P16,L12-14; P18, L14-18) and can be observed in fig. 12.

Katja Fennel
Editor

**Comments from reviewer 1:**

P6L5: "transparency"
AR: Done (P6, L18).

P12L9: section title suggestion: Nutrient load reduction scenario
AR: Done.

P12L21 (and everywhere else in the manuscript, there are many occurrences): change "with" to "by"
AR: Done.

P12L22: replace "to" by "in"
AR: This paragraph is re-written and the comment is therefore not valid any more.

P16L1-3: change to something like "Permanent retention was relatively stable during the simulated period, while fluctuations in the temporal retention reflects the effect of varying riverine nutrient input (Fig. 10c,d).".
AR: Done (P17, L4-5).

Also, temporal retention of P has increased, Can you comment on that?
AR: Yes, this is now mentioned in the section 3.2 (P18, 17-20).

P17L11-16: This new part could be polish for clarity.
AR: This section is now revised.

P17L14: change "has" to "have"
AR: Done.

P18L23: change "parameters" to "processes"
AR: The whole paragraph is revised, section 3.2.2 (P19).

P18L26: change "relation" to "relationship"
AR: Done (P19, L27).

P18L27-28: replace to "… the filter efficiency. Nixon et al (1996) show that including the depth in the analysis of retention"
AR: Done, P1928-30.

P19L1-31: This added text could be improved.
P19L1-4: suggested replacement for the first 2 sentences: "In the model, denitrification is an O2 dependent process that has a maximum rate at O2 concentration…".
AR: Done, but parts of this section is also moved to the new method section 2.4.1 (P12-13).

P19L5-9: Are you talking about the model here or in general? This is not described in the Methods so we don't know if this relates to your results or not.
AR: It is mentioned in the method section 2.2.2 that the release of phosphate is redox dependent (P8, L18-20) and it is now also explained in the section 2.4.1 (P12-13).

P19L9-12: What do you mean by manually? This is vague, you need to provides details on what you did. It could be mentioned in the Methods.
AR: This is now rewritten and described in the new section 2.4.1 in the revised manuscript (P12-13).

P19L17: Is this due only to denitrification? what about higher primary production due to more P available? Did you compare the different effects and found that denitrification is the dominant process You mention this later in the load reduction experiment.?
AR: The permanent retention of N increased and the proportion of the N that was denitrified also increased. The permanent retention, burial, of P decreased. Thus there was a change in the N:P ratio. However, the temporary retention of N did not change significantly, but there was a decreased export of N to the open sea, while the P export increased compared to the original run. This paragraph is re-formulated (P20, L14-30)

P19L17-25: Suggested changes for clarity: "Denitrification increased in magnitude and as a proportion of the permanent retention from 92% to 94%, while the proportion of burial decreased . The decreased in P retention was due to higher released of P from the sediment. Denitrification has been shown to increase in areas with longer residence times (Nixon et al., 1996, Nielsen et al., 2001, Finlay et al., 2013). In the simulation the residence time was only six days on average and the filter efficiencies of N and P were estimated to be 10% and 9%, respectively, which were lower than in the Stockholm Archipelago…"
AR: Thank you for the suggested changes, however, large parts of this paragraph is re-formulated (P20, L14-30)

Also, I don't see why the results L23-25 are included here. Can you be more explicit about the comparison with your system?
AR: Denitrification is one of the removal processes, permanent retention, which is discussed here. The magnitude of the denitrification has been shown to be increase with areas having longer residence time compared to shorter. The result in the study in the Randers fjord is here compared to the Stockholm archipelago where we have both longer residence time and more retention due to denitrification. This is reformulated in the revised manuscript (P20, L14-19).

P19L29-31: This is true everywhere. Please clarify your statement.
AR: This sentence is now rewritten (P20, L20-22).

P19-20L33-4: Algae uptake changes the pool of nutrients, please rephrase to something like this: Assimilation of nutrients by benthic algae does not directly change the inventory of N and P but transfers inorganic N and P into organic material..."
AR: This sentence are now rewritten (P20-21, L32-1).

What is the expected effect on retention? You need to relate this to your results, this is part of the uncertainty of your results.
AR: A discussion about this is now added (P21, L3-12).

P20L13: suggested change to title: "Response to nutrient load reduction.
AR: Done (P21, L21).

P21L11: suggested change: "is" instead of "becomes"
AR: Done (P22, L19).

P21L14: Change "From these experiments it seems that" to "These results indicate that".
AR: Done (P22, L22).

P21L17-18: suggested change: "… concentrations might be expected locally. However, this effect largely depends on the water residence time and on which nutrient limits the seasonal phytoplankton production initially…"
AR: Done (P22, L24-26).

P21L20-29: I don't see the use of this paragraph.
AR: We agree, and large part of the paragraph is now removed.

P21L22-25: not sure what that means, please clarify.
AR: These sentences are now rewritten (P22, L28-30).

P23L6: replace "to" with "in".
AR: Done (P24, L6).

P23L8-10: is that permanent or temporary? Can you indicate their ratio?
AR: This retention is the total, thus a sum of permanent and temporary retention (defined in section 2.4), which now is clarified in the conclusions (P24, L9). The temporary retention in this case is negative, thus we have a decrease in the inventory of N and P. The permanent retention is larger about 98 %. In the conclusions we would prefer to give the total number of retention, however, the different values are given in fig. 9 from which the different ratios can be calculated.

P25L1: journal names should be abbreviated
AR: Done (P26-32).

P25L16-18: add database title: "Baltic Environmental Database" and use the short url provided: http://www.balticnest.org/bed
AR: Done (P26, L16).

P44: is there a reason to not start the x-axis at 0? It seems odd.
AR: No, there is no reason and we also took the opportunity to remake the figure. In the new figure (fig. 14) we think it is easier to compare the filter efficiencies of N and P.

**Additional changes by the authors:**

"- a model study" has been removed from the title, since the word "Modelling" should be enough.

The keyword "nutrient retention " is already in the title and is therefore changed to "denitrification" .

[revised manuscript text omitted]